# Terahertz-field-induced polar charge order in electronic-type dielectrics

H. Yamakawa[1], T. Miyamoto[1✉], T. Morimoto[1], N. Takamura[1], S. Liang [1], H. Yoshimochi[2], T. Terashige[3], N. Kida[1], M. Suda [4,6], H. M. Yamamoto [4], H. Mori[5], K. Miyagawa[2], K. Kanoda [2] & H. Okamoto [1,3✉]

Ultrafast electronic-phase change in solids by light, called photoinduced phase transition, is a central issue in the field of non-equilibrium quantum physics, which has been developed very recently. In most of those phenomena, charge or spin orders in an original phase are melted by photocarrier generations, while an ordered state is usually difficult to be created from a non-ordered state by a photoexcitation. Here, we demonstrate that a strong terahertz electric-field pulse changes a Mott insulator of an organic molecular compound in $\kappa$-(ET)$_2$Cu [N(CN)$_2$]Cl (ET = bis(ethylenedithio)tetrathiafulvalene), to a macroscopically polarized charge-order state; herein, electronic ferroelectricity is induced by the collective inter-molecular charge transfers in each dimer. In contrast, in an isostructural compound, $\kappa$-(ET)$_2$Cu$_2$(CN)$_3$, which shows the spin-liquid state at low temperatures, a similar polar charge order is not stabilized by the same terahertz pulse. From the comparative studies of terahertz-field-induced second-harmonic-generation and reflectivity changes in the two compounds, we suggest the possibility that a coupling of charge and spin degrees of freedom would play important roles in the stabilization of polar charge order.

[1] Department of Advanced Materials Science, University of Tokyo, Chiba 277-8561, Japan. [2] Department of Applied Physics, University of Tokyo, Bunkyo-Ku 113-8656, Japan. [3] AIST-UTokyo Advanced Operand-Measurement Technology Open Innovation Laboratory, National Institute of Advanced Industrial Science and Technology, Chiba 277-8589, Japan. [4] Division of Functional Molecular Systems, Research Center of Integrative Molecular Systems (CIMoS), Institute for Molecular Science, Okazaki 444-8585, Japan. [5] Institute for Solid State Physics, University of Tokyo, Chiba 277-8581, Japan. [6]Present address: Department of Molecular Engineering, Kyoto University, Kyoto 615-8510, Japan. ✉email: miyamoto@k.u-tokyo.ac.jp; okamotoh@k.u-tokyo.ac.jp

Photoinduced phase transitions in which electronic structures are changed in the sub-picosecond time scale are attracting much attention not only as a recent topic in solid-state physics but also as a possible mechanism for ultrafast optical switching functions. In most of those phenomena, charge or spin orders in an original phase are melted by photocarrier generations; typical examples are photo-induced insulator to metal transitions observed in transition metal compounds[1,2] and organic molecular materials[3]. It is generally difficult to create an ordered state from an original less-ordered one by a photo-excitation[4]. To overcome this difficulty, the application of a strong terahertz pulse is an effective way. In fact, it was reported that a macroscopic polarization was generated by an electric-field component of a terahertz pulse in an organic molecular compound, tetrathiafulvalene-p-chloranil, in the paraelectric phase[5] and a transition metal oxide $SrTiO_3$ in the quantum paraelectric phase[6]. In addition, a polarization control of $SrTiO_3$ was successfully made by exciting a specific lattice vibration with a mid-infrared (IR) pulse[7].

Here, we focus on the creation of a macroscopically polar state by an irradiation of a nearly monocyclic terahertz pulse via electric-field-induced phase transition. The studied materials are Mott insulators of ET-based organic molecular compounds (ET = bis(ethylenedithio)tetrathiafulvalene) expressed by κ-(ET)$_2$X (X: anions). These compounds include charge degree of freedom on each site and are likely to exhibit instability to charge order (CO). In this regard, the possibilities of CO formations and related extraordinary dielectric properties such as electronic ferroelectricity or polar CO[8–11] and dipolar liquid[12] are being studied extensively. The previous theoretical studies suggested that it depends on the inter-site Coulomb interaction whether the polar CO is stabilized or not in these compounds[13,14]. The electron-lattice interaction might also be important on the stabilization of the polar CO[15]. In fact, molecular displacements stabilize the CO in another ET-based molecular compound, α-(ET)$_2$I$_3$ (ref. [16]).

In the present study, we aimed at creating polar CO states in the κ-(ET)$_2$X compounds by using a strong terahertz pulse as an external stimulus[17–19]. The studied materials are κ-(ET)$_2$Cu[N(CN)$_2$]Cl[20] and κ-(ET)$_2$Cu$_2$(CN)$_3$[21]; hereafter, these are abbreviated as κ-Cl and κ-CN, respectively. In κ-Cl and κ-CN, ET molecules form layer structures as shown in Fig. 1a[22,23]. On a layer, ET dimers form a distorted triangular lattice (Fig. 1b). In each dimer, the molecular orbitals of two ETs are hybridized via the intradimer transfer integral $t_1$ and split into the bonding and anti-bonding orbitals; these form two-dimensional (2D) bands via the interdimer transfer integrals $t$ and $t'$ (Fig. 1b). The nominal valence of ET is +0.5 so that a hole occupies the anti-bonding orbital on each dimer, and the band is half-filled. Because the Coulomb repulsion energy $U_{dimer}$ on each dimer approximated by $2t_1$ is larger than the bandwidth, both the compounds are Mott insulators[24].

In κ-Cl, the spin system exhibits an antiferromagnetic order (AFO) below $T_c = 27$ K[25], whereas, in κ-CN, no magnetic order appears above 32 mK[26]. This difference is considered to arise from the degrees of magnetic frustrations in the triangular spin arrangements, which are characterized by $t'/t$; as indicated in Fig. 1c, $t'/t$ (= 0.83–0.99) in κ-CN is approximately equal to one, whereas $t'/t$ (= 0.44–0.52) in κ-Cl is significantly smaller[27,28]. In addition, previous theoretical studies[13,14,29–31] and dielectric measurements[8,9] have reported that they include the instability to CO, wherein charges in each dimer are disproportionate. In κ-Cl, the dielectric constant exhibits Curie–Weiss behavior with decreasing temperature down to $T_c$; below this, a hysteresis loop is observed in the polarization-electric-field curve[9]. These results indicate that κ-Cl is in a ferroelectric CO phase below $T_c$. Such simultaneous appearances of CO and AFO are reminiscent of a coupling of charge and spin degrees-of-freedom[9,11]. However, from the molecular vibrational spectroscopy, the charge disproportionation appears to be highly marginal (<0.01)[32]. Thus, the formation of ferroelectric CO is under debate.

In this paper, we report that a macroscopically polar state is generated by a terahertz electric field in κ-Cl via the electric-field-induced Mott-insulator to CO transition. From the comparative studies of terahertz-pump optical-probe spectroscopy on κ-Cl and κ-CN, we clarify the roles of the intermolecular Coulomb interaction, the electron–lattice interaction, and the magnetic interaction on the stabilization of the polar CO in κ-Cl under strong electric fields. These findings provide great physical insights into the Mott physics in organic molecular materials.

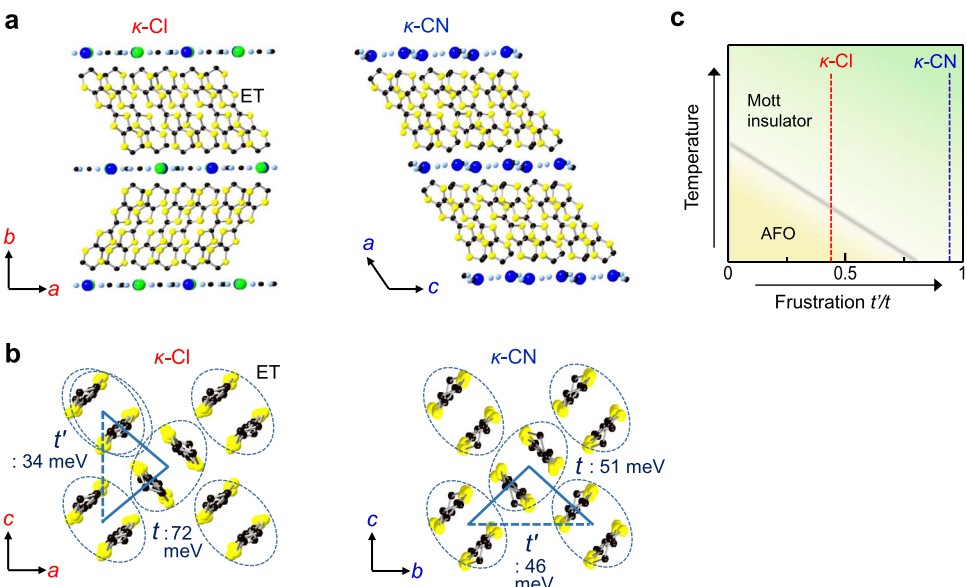

**Fig. 1 Crystal structures and the phase diagram of κ-Cl and κ-CN. a** The crystal structures. **b** Schematics of molecular arrangements on the 2D molecular planes. $t$ and $t'$ are interdimer transfer integrals[27]. **c** The conceptual phase diagram.

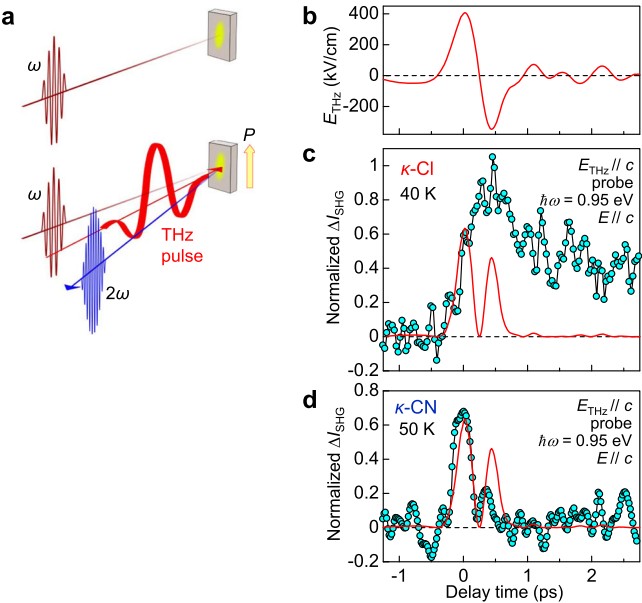

**Fig. 2 Terahertz-pulse-pump SHG-probe measurements. a** Schematic of the experimental setup. **b** Terahertz electric-field waveform $E_{THz}(t_d)$. **c**, **d** Time evolutions of electric-field-induced SHG signals $\Delta I_{SHG}(t_d)$ in **c** $\kappa$-Cl and **d** $\kappa$-CN. The red lines in the lower two panels show time characteristics of $[E_{THz}(t_d)]^2$. The standard deviation of the normalized $\Delta I_{SHG}$ data is ±0.1.

## Results

**Transient second harmonic generation by a terahertz electric field**. First, we report the terahertz-pulse-pump second-harmonic-generation (SHG)-probe measurement as illustrated in Fig. 2a. In the SHG measurements, we set the photon energy of the incident pulse to be 0.95 eV, since the absorption depth at 0.95 eV is long enough as compared to the coherence length of the reflection-type SHG in $\kappa$-Cl and $\kappa$-CN. In addition, the detailed SHG measurements were previously performed with the same energy of the incident pulse on the ET compound, $\alpha$-(ET)$_2$I$_3$, in which the polar CO state is stabilized below $T_c = $ 135 K, and the SHG signals were observed in the steady state[16,33]. In both $\kappa$-Cl and $\kappa$-CN, no SHG signals were detected in the steady states at all the temperatures from 294 to 10 K irrespective of the polarization direction of the incident pulse in contrast to $\alpha$-(ET)$_2$I$_3$. This indicates that no macroscopically polar state is formed in $\kappa$-Cl even below $T_c = 27$ K as well as in $\kappa$-CN. The nature of the electronic state below $T_c = 27$ K in $\kappa$-Cl will be discussed later. To investigate the possibility of the formation of the terahertz electric field-induced polar CO, we performed the terahertz pulse-pump SHG-probe measurement on $\kappa$-Cl at 40 K and on $\kappa$-CN, at 50 K. These temperatures are enough above the critical temperature $T_c = 27$ K in $\kappa$-Cl.

Figure 2b shows the electric-field waveform $E_{THz}(t_d)$ of the terahertz pulse. Its amplitude $E_{THz}(0)$ is ~400 kV/cm. When this electric field is applied along the $c$-axis on the $ac$ plane in $\kappa$-Cl, an SHG signal $\Delta I_{SHG}(t_d)$ appears as shown in Fig. 2c. The electric fields $E$ of both the incident probe light and second-harmonic light are also polarized parallel to the $c$-axis. $\Delta I_{SHG}(t_d)$ increases in accord to $[E_{THz}(t_d)]^2$ (the red line in Fig. 2c) without delay, suggesting that the initial response is electronic in nature. $\Delta I_{SHG}(t_d)$ remains after the electric field diminishes. The important feature is that the SHG signal does not decrease so much at the time when the terahertz electric field crosses zero and changes its sign at ~0.2 ps. This observation suggests that the polar state formed by the first positive peak of the terahertz electric field around the time origin is stabilized within ~0.3 ps

and its polarization does not reverse by the negative electric fields from 0.25 to 0.75 ps. Considering the instability to the CO with a charge disproportionation in each dimer previously reported in $\kappa$-type ET compounds[8–12], we tentatively assign the observed SHG to the polar CO in which charge disproportionation is aligned along the electric field direction.

Figure 2d shows the time characteristic of the SHG signal with the terahertz electric field (Fig. 2b) parallel to the $c$-axis on the $bc$ plane in $\kappa$-CN at 50 K. The electric fields $E$ of both the incident probe light and second-harmonic light are also polarized parallel to the $c$-axis. The SHG signal shows a sharp peak around the time origin and then rapidly decreases to zero at the time when the electric field crosses zero. After that, the SHG signal increases again under the presence of the negative electric field from 0.25 to 0.75 ps. In this case, it is natural to consider that the polarization reverses depending on the electric field direction. The SHG signal vanishes immediately after $E_{THz}(t_d)$ diminishes for $t_d > 0.75$ ps. The signal is roughly proportional to $[E_{THz}(t_d)]^2$ in all the time region so that it can be regarded as a kind of third-order optical nonlinearity[34]. Thus, in $\kappa$-CN, the polarization induced by an electric field is not stabilized. This is in contrast to the case of $\kappa$-Cl. From the comparison of the magnitude of the observed SHG signal in $\kappa$-Cl ($\kappa$-CN) with that of $\alpha$-(ET)$_2$I$_3$, the maximum of the transient polarization in $\kappa$-Cl ($\kappa$-CN) is roughly estimated to be 10% (8%) of the polarization (~1 μC/cm$^2$) in the ferroelectric CO phase of $\alpha$-(ET)$_2$I$_3$. The experimental conditions of the SHG measurements, the framework of the electric-field-induced SHG as the third-order optical nonlinearity, and the estimations of the electric-field-induced polarizations in $\kappa$-Cl and $\kappa$-CN are reported in Supplementary Note 1.

**Transient reflectivity changes by a terahertz electric field in $\kappa$-Cl: electric-field dependence**. To obtain information about the electric-field-induced polar state in $\kappa$-Cl, we performed terahertz-pulse-pump optical-reflectivity-probe spectroscopy. Figure 3a, b shows the reflectivity $R$ and optical-conductivity $\sigma$ spectra in $\kappa$-Cl at 40 K with the electric field of lights $E$ parallel to the $c$-axis ($E$//$c$). The broad peak at approximately 0.2 eV is attributed to the interdimer or Mott-gap transition[35] (the green arrow in Fig. 3c). The peak at approximately 0.43 eV is assigned to the intradimer transition between the bonding and antibonding orbitals[35] (the orange arrow in Fig. 3c). The energies of two transitions increase with the increase of $U_{dimer}$, while the interdimer transition is more sensitive to the interdimer transfer integral or equivalently the bandwidth $\delta$. The $R$ and $\sigma$ spectra are approximately reproduced by the sum of two electronic transitions indicated by the green and orange lines and three sharp phonon absorptions, as shown by the red lines in Fig. 3a, b. The details of these analyses are reported in Supplementary Note 2.

Figure 3e shows the time evolutions of reflectivity changes $\Delta R(t_d)/R$ at three probe energies, 0.108, 0.33, and 0.5 eV, together with the electric-field waveform $E_{THz}(t_d)$ of the terahertz pulse. The terahertz electric field is parallel to the $c$-axis ($E_{THz}$//$c$) and the maximum electric field is 288 kV/cm ($E_{THz}(0) = 288$ kV/cm). The electric field of the probe pulse is also parallel to the $c$-axis ($E$//$c$). At 0.5 and 0.33 eV, $\Delta R(t_d)/R$ decreases and recovers with a few picoseconds; meanwhile, at 0.108 eV, $\Delta R(t_d)/R$ is initially negative and changes sign at approximately 0.2 ps. The $|\Delta R(t_d)/R|$ values are maximized at $t_d$ ~0.5 ps, around which the electronic-state change is completed. In addition, an oscillation of period ~1 ps is superimposed on $\Delta R(t_d)/R$; this is discussed subsequently.

The magnitudes of $\Delta R(t_d = 0 \text{ ps})$ are plotted in Fig. 3f; it exhibits a characteristic minus-plus-minus structure. By assuming a blue shift (1.8 meV) and an intensity decrease (2.9%) of the intradimer transition, and a blue shift (6.1 meV) of the interdimer

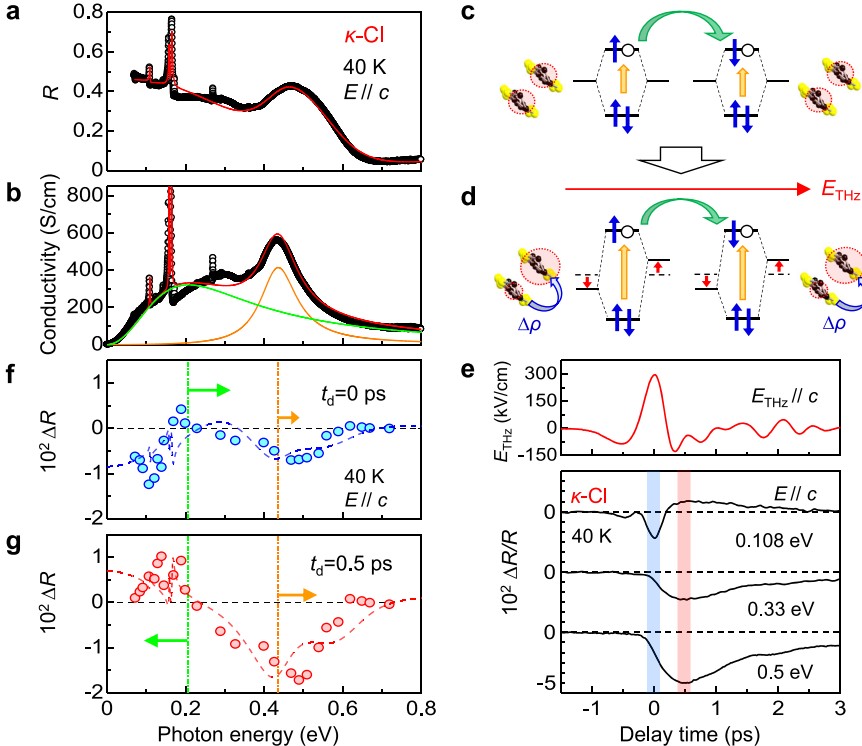

**Fig. 3 Electric-field-induced changes of polarized reflectivity spectra and deduced charge and molecular dynamics in $\kappa$-Cl. a, b** Steady-state **a** optical-reflectivity $R$ and **b** optical-conductivity $\sigma$ spectra. **c, d** Schematic of hybridized molecular orbitals in each dimer **c** without and **d** with electric fields. Blue arrows and open circles show spins of electrons and holes, respectively. Intradimer and interdimer transitions are indicated by orange and green arrows, respectively. **e** Terahertz electric-field waveform, $E_{THz}(t_d)$, and time evolutions of reflectivity changes, $\Delta R(t_d)/R$. **f, g** Spectra of reflectivity changes **f** $\Delta R(0\,\text{ps})$ and **g** $\Delta R(0.5\,\text{ps})$. The error bars are smaller than the data points.

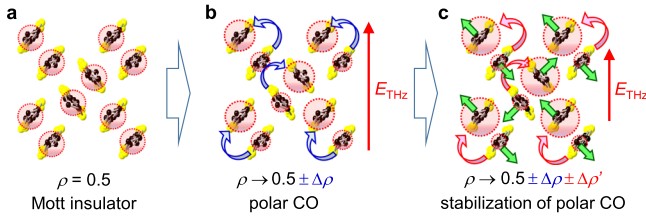

**Fig. 4 Possible charge and molecular dynamics induced by the terahertz electric field. a** Mott insulator state, **b** polar CO state induced by the terahertz electric field, and **c** polar CO state stabilized by molecular displacements decreasing the dimerization. The blue and red curved arrows indicate the intradimer charge transfers and the green arrows indicate the molecular displacements. The polar CO state is stabilized by the decrease of the dimerization only in $\kappa$-Cl.

transition, we can approximately reproduce the $\Delta R(t_d = 0\,\text{ps})$ spectrum, as shown by the blue broken line in Fig. 3f. The directions of the shifts of the two bands are shown by the arrows in the same figure. Those spectral changes are explained as follows (see Supplementary Note 2 for details). The electric field induces the site energy difference in each dimer. This change gives rise to the charge disproportionation along the electric-field direction (Fig. 3d and Fig. 4a, b) and also increases the splitting of the bonding and anti-bonding orbitals, which causes the blue shift and the intensity decrease of the intradimer transition.

Figure 3g shows the $\Delta R$ spectrum at $t_d = 0.5\,\text{ps}$, $\Delta R(t_d = 0.5\,\text{ps})$, in which $\Delta R$ below 0.2 eV rather increases. By assuming a blue shift (4.0 meV) and a further intensity decrease (5.4%) of the intradimer transition, and a red shift (5.7 meV) of the interdimer transition, we can approximately reproduce the $\Delta R(t_d = 0.5\,\text{ps})$

spectrum as shown by the red broken line in Fig. 3g. The directions of the shifts of the two bands are also shown by the arrows in the same figure. A possible explanation of such a delayed response is the decrease in the dimerization in each dimer as illustrated in Fig. 4b, c. The electric field not only induces the charge transfer in each dimer but also pull apart charge-disproportionated two molecules. The theoretical studies indicate that the decrease in $t_1$ in each dimer favors the CO so that those molecular motions would make the polar CO more stabilized and its lifetime longer. The decrease in $t_1$ in each dimer tends to decrease the splitting of bonding- and anti-bonding orbitals, while the stabilization of the CO means that the site-energy difference between two molecules in each dimer is enhanced, which tends to increase the splitting of bonding- and anti-bonding orbitals. The experimental result shows the blue shift of the intradimer transition so that the latter effect overcomes the former effect. In this case, both the decrease in $t_1$ and the increase in the orbital splitting should suppress the oscillator strength of the intradimer transition. At $t_d = 0.5\,\text{ps}$ the interdimer-transition rather shows a red shift. The blue shift of the intramolecular transition suggests the increase in the splitting of the bonding and anti-bonding orbitals corresponding to $U_{dimer}$, which cannot explain the red shift of the interdimer transition. A possible origin for the red shift is the increase in the bandwidth by the increase in $t$ and $t'$ through the molecular motions corresponding to the release of the dimerization, which is also detailed in Supplementary Note 2.

In $\kappa$-Cl at 40 K, the time evolution of $\Delta I_{SHG}(t_d)$ (Fig. 2c) reflecting the electric-field-induced polarization and that of $\Delta R(t_d)/R$ at 0.5 eV (Fig. 3e) associated with the electric-field-induced charge disproportionation in each dimer are almost in agreement with each other (see Supplementary Note 3). This shows that the dynamics of the electric-field-induced polar CO

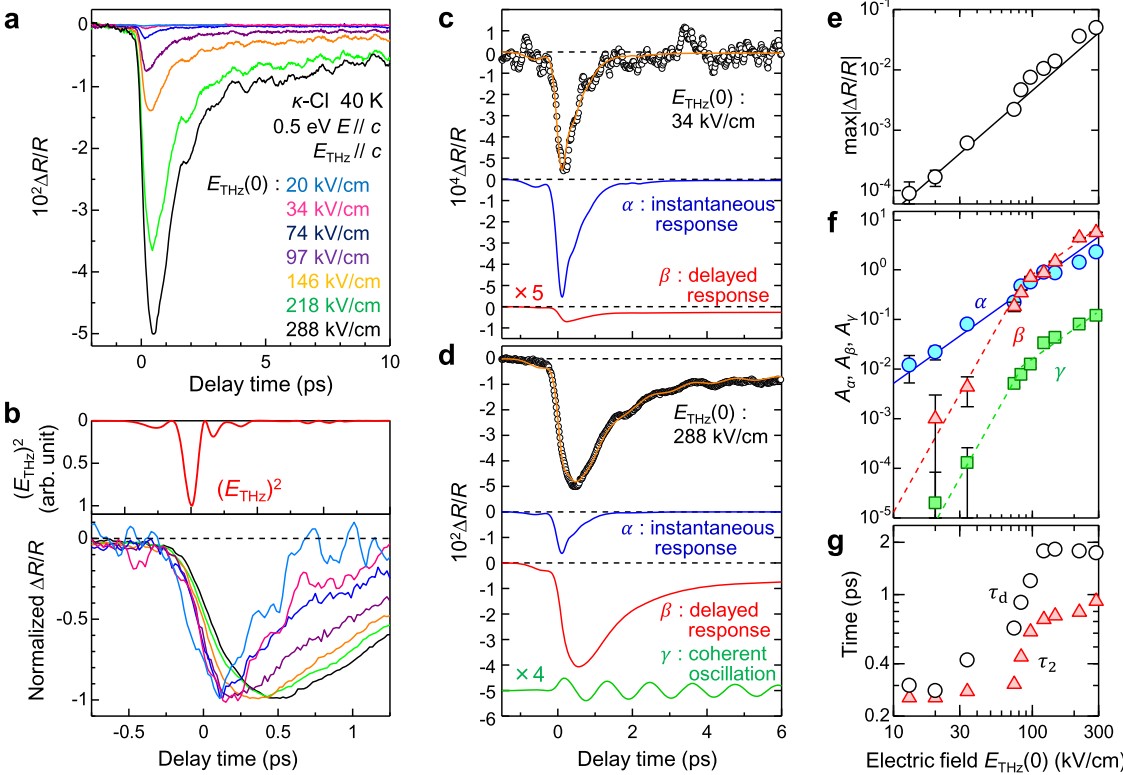

**Fig. 5 Time evolutions of reflectivity changes induced by the terahertz electric field and their analyses in κ-Cl. a** Time evolutions of reflectivity changes $\Delta R(t_d)/R$ at 0.5 eV and at 40 K. **b** The terahertz electric-field waveform $[E_{THz}(t_d)]^2$ (red line in the upper panel) and time evolutions of normalized $\Delta R(t_d)/R$ at 0.5 eV (the lower panel). **c, d** Typical time evolutions of $\Delta R(t_d)/R$ and their fitting analyses; **c** $E_{THz}(0) = 34$ kV/cm and **d** $E_{THz}(0) = 288$ kV/cm. The orange lines show the fitting curves. The standard deviation of the $\Delta R/R$ data is $\pm 5 \times 10^{-5}$. **e–g** Terahertz-electric-field $[E_{THz}(0)]$ dependences of parameters; **e** max$|\Delta R(t_d)/R|$. The error bars indicate the standard deviation. **f** Amplitudes of the three terms in Eq. (1). $A_\alpha$: instantaneous responses, $A_\beta$: delayed responses, and $A_\gamma$: coherent oscillations. The error bars indicate the standard deviation of the least-squares fit. **g** The decay time $\tau_2$ in the second term in Eq. (1) (red triangles) and the time $\tau_d$ until $\Delta R(t_d)/R$ decreases to the half of max$|\Delta R(t_d)/R|$ (open circles).

at 40 K can be discussed from the time evolutions of $\Delta R(t_d)/R$ at 0.5 eV.

To clarify the time characteristic of charge and molecular dynamics in the formation of CO by the electric field, we measured the dependence of the time evolutions of $\Delta R(t_d)/R$ on the electric-field amplitudes. In Fig. 5a, we show the time evolutions of $\Delta R(t_d)/R$ at 0.5 eV and at 40 K for various electric-field amplitudes $E_{THz}(0)$ with $E_{THz}(t_d)//c$. The probe pulse is polarized along the $c$-axis ($E//c$). Figure 5b shows the same $\Delta R(t_d)/R$ data in the normalized scale to clearly illustrate the difference in the dynamics. The red solid line in the upper panel represents the square of the terahertz electric-field waveform, $[E_{THz}(t_d)]^2$. When the electric-field amplitude is small, e.g., $E_{THz}(0) = 20$ kV/cm (the light-blue line), the reflectivity initially decreases following $[E_{THz}(t_d)]^2$ and it shows an additional small decrease within 0.2 ps. After that, it recovers with a decay time of approximately 0.3 ps. As the electric-field amplitude increases, both of the initial decreases of the reflectivity and its recovery slow down. For $E_{THz}(0)$ larger than 100 kV/cm, the component with the longer decay time and the coherent oscillation appear.

As mentioned above, the results of the $\Delta R(t_d)/R$ spectra and their analyses for the case with the strong electric field ($E_{THz}(0) = 288$ kV/cm) indicate that the dynamics consist of three processes; the initial instantaneous decrease in the reflectivity due to the intradimer charge transfers by the terahertz electric field (process $\alpha$), the delayed response due to the stabilization of the electric-field-induced polar state by the weakening of the dimerization by the molecular displacements, i.e., the stabilization of the polar CO state (process $\beta$), and the subsequent coherent oscillation (process $\gamma$). To

reproduce the time evolutions of $\Delta R(t_d)/R$ reflecting these three processes, we adopted the following function:

$$\Delta R/R(t) = \int_{-\infty}^{t} [E_{THz}(t')]^2 F(t - t') dt'$$

$$F(t) = A_\alpha \exp\left(-\frac{t}{\tau_1}\right) + A_\beta \left[1 - \exp\left(-\frac{t}{\tau_r}\right)\right] \left[\delta \exp\left(-\frac{t}{\tau_2}\right) + (1 - \delta) \exp\left(-\frac{t}{\tau_3}\right)\right]$$

$$+ A_\gamma \exp\left(-\frac{t}{\tau_{OSC}}\right) \cos(\omega_{OSC} t + \phi_{OSC})$$

(1)

The first term in $F(t)$ is the instantaneous response, which increases with $[E_{THz}(t_d)]^2$ and decays with the decay time $\tau_1$. This term represents process $\alpha$. The second term is the delayed response increasing with the rise time $\tau_r$, which represents process $\beta$. As seen in Fig. 5a, the signals do not exhibit a simple single exponential decay and include the components remaining for $t_d > 10$ ps; therefore, we introduce two decay times $\tau_2$ and $\tau_3$. $\delta$ is the parameter determining the ratio of these two components. In our model, the charge disproportionate states initially produced via process $\alpha$ are stabilized by the molecular displacements via process $\beta$. The results shown in Fig. 5b make us expect that the degree of the stabilization of the initial charge disproportionate states changes depending on the electric field amplitudes or the magnitudes of the initial charge disproportionation themselves, and dominates both the rise time and the decay time of the signals. An appropriate way for the simulation of such an electric-field dependence is to express processes $\alpha$ and $\beta$ by two terms separately as Eq. (1), in which we consider process $\beta$ as the formation of another CO state instead of the initial charge disproportionation state. In other words, the stabilization

of the initial charge disproportionate state is expressed by the exchange of the first term with the second term. The third term represents the coherent oscillation (process $\gamma$). $\omega_{OSC}$, $\tau_{OSC}$, and $\phi_{OSC}$ are the frequency, decay time, and phase, respectively, of the oscillation.

In Fig. 5c, d, we show the results of the analyses of $\Delta R(t_d)/R$ for $E_{THz}(0) = 34$ kV/cm and 288 kV/cm as examples of the weak and strong electric-field cases, respectively. The time evolutions of $\Delta R(t_d)/R$ shown by the open circles are well reproduced by Eq. (1) as shown by the orange lines. The parameter values with error bars (one standard deviation) are $\tau_1 = 0.30 \pm 0.02$ ps, $\tau_2 = 0.28 \pm 0.02$ ps, $\tau_3 = 27 \pm 1$ ps, $\tau_r = 0.28 \pm 0.02$ ps, $\delta = 0.92 \pm 0.02$, $\omega_{OSC} = 29.1 \pm 0.3$ cm$^{-1}$, $\phi_{OSC} = 57 \pm 5°$ and $\tau_{OSC} = 11 \pm 0.6$ ps for $E_{THz}(0) = 34$ kV/cm and $\tau_1 = 0.30 \pm 0.02$ ps, $\tau_2 = 0.92 \pm 0.01$ ps, $\tau_3 = 27 \pm 1.5$ ps, $\tau_r = 0.28 \pm 0.02$ ps, $\delta = 0.9 \pm 0.02$, $\omega_{OSC} = 29.1 \pm 0.3$ cm$^{-1}$, $\phi_{OSC} = 57 \pm 5°$, and $\tau_{OSC} = 8.3 \pm 0.6$ ps for $E_{THz}(0) = 288$ kV/cm. Each component is also shown in the lower part of the respective figures. The third term for $E_{THz}(0) = 34$ kV/cm was not shown, since it was very small.

The generation mechanism of coherent oscillation can be explained as follows. Since strong IR-active phonons are absent around 29 cm$^{-1}$ (Supplementary Note 4), the direct phonon excitation by the terahertz pulse is not the origin of the coherent oscillation. We can consider that this oscillation is driven by the molecular displacements induced via the stabilization of the charge disproportionation state (process $\beta$). The frequency 29.1 cm$^{-1}$ corresponds to the period 1.15 ps, which is four times as large as $\tau_r = 0.28$ ps. This is reasonable because $\tau_r$ should be dominated by the time required for the molecular displacements to occur, which can be approximated as a quarter of the oscillation period. A similar coherent oscillation is also observed in the photoinduced insulator-metal transition in a similar $\kappa$-type ET compound;[36,37] this is interpreted as an oscillation modulating the intradimer molecular distance, i.e., a dimer mode.

This mechanism of the generation of the coherent oscillation is a kind of displacive excitation of a coherent phonon (DECP)[38]. An initial phase of a coherent oscillation by the DECP mechanism is generally 0 or $\pi$. In our results, however, the initial phase $\phi_{OSC}$ is 57°. This is explained in the following way. The molecular displacements responsible for the observed oscillation are caused by the charge disproportionation, which is induced by the terahertz electric field. This charge disproportionation occurs as shown by the blue lines in Fig. 5c, d. The center of the gravity of this signal is not located at the time origin but located at approximately 0.2 ps. The period of the oscillation is approximately 1.15 ps, so that 0.2 ps corresponds to approximately 60°, which is almost equal to the evaluated initial phase.

Similar analyses were performed on the other $\Delta R(t_d)/R$ data measured with various electric field amplitudes $E_{THz}(0)$. In those analyses, $\tau_1 = 0.30$ ps, $\tau_3 = 27$ ps, $\tau_r = 0.28$ ps, $\omega_{OSC} = 29.1$ cm$^{-1}$, and $\phi_{OSC} = 57°$ are fixed. We show the electric-field dependences of the maximum values of $|\Delta R(t_d)/R|$, max$|\Delta R(t_d)/R|$, in Fig. 5e, the values of $A_\alpha$, $A_\beta$, and $A_\gamma$ which characterize the magnitudes of three processes $\alpha$–$\gamma$ in Fig. 5f, and the values of $\tau_2$ which dominate the initial decay of the electric-field-induced polar state in Fig. 5g. More precisely, the decay dynamics is reproduced by the sum of two exponential terms with the shorter decay time $\tau_2$ and the longer decay time $\tau_3$. In order to understand the electric-field dependence of the decay time of the polar state more clearly, we defined the time until the $|\Delta R(t_d)/R|$ signal becomes half the max$|\Delta R(t_d)/R|$ as the effective decay time $\tau_d$ and plotted it in Fig. 5g.

For $E_{THz}(0) < 40$ kV/cm, max$|\Delta R(t_d)/R|$ is proportional to $[E_{THz}(0)]^2$ (Fig. 5e), and the amplitudes of the delayed component $A_\beta$ and the coherent oscillation $A_\gamma$ are very small (Fig. 5f). $\tau_2$ ($\tau_d$) is relatively small, being approximately 0.3 ps.

These results indicate that the polar CO is not stabilized so much for $E_{THz}(0) < 40$ kV/cm. For $E_{THz}(0) > 70$ kV/cm, in contrast, max$|\Delta R(t_d)/R|$ shows a further nonlinear behavior (Fig. 5e), and $A_\beta$ and $A_\gamma$ increase in a threshold manner (Fig. 5f). $\tau_2$ ($\tau_d$) becomes very long, being 0.8–1 ps ($\sim$2 ps) (Fig. 5g). These results indicate that a threshold electric field, which is estimated to be approximately 80 kV/cm, exists for the structural changes (or the molecular displacements weakening the dimerization). Above this electric field, the polar CO state is more stabilized and its lifetime becomes relatively long. This suggests that a small but finite potential barrier would be produced between the paraelectric Mott insulator state and the polar CO state.

**Transient reflectivity changes by a terahertz electric field in $\kappa$-Cl: temperature dependence**. In this section, we present the results of the terahertz-electric-field-induced reflectivity changes $\Delta R(t_d)/R$ at various temperatures in $\kappa$-Cl and discuss the temperature dependence of the responses to the electric field. Figure 6a, b shows the time evolutions of $\Delta R(t_d)/R$ at 0.5 eV for $E_{THz}//c$ ($E_{THz}(0) = 288$ kV/cm) and $E//c$ above and below 40 K, respectively. In Fig. 6c–e, we show the normalized $\Delta R(t_d)/R$ together with the square of the terahertz field, $[E_{THz}(t_d)]^2$. The temperature dependence of the maximum values of $|\Delta R(t_d)/R|$, max$|\Delta R(t_d)/R|$, is shown in Fig. 7a. With the decrease in the temperature from 290 K, max$|\Delta R(t_d)/R|$ increases; it attains the maximum value at 40 K. Simultaneously, the rise time of $|\Delta R(t_d)/R|$ increases and the decay time of $|\Delta R(t_d)/R|$ is prolonged as seen in Fig. 6d. This suggests that the $\beta$ component reflecting the stabilization process of the polar CO state is enhanced with the decrease in the temperature. In fact, the time characteristic of $\Delta R(t_d)/R$ at 290 K shown in Fig. 6d is very similar to that of the $\alpha$ component of $\Delta R(t_d)/R$ at 40 K shown in Fig. 5d and the $\beta$ component does not exist at 290 K. All these results indicate that at approximately 40 K, the instability to the polar CO state is maximized and the electric-field-induced CO is most likely to be stabilized. By using the value of $\tau_d$ obtained from the decay dynamics of $\Delta R(t_d)/R$, we can see that the lifetime of the polar CO state increases with the decrease of temperature down to 40 K, e.g., $\tau_d \sim 0.44$ ps at 290 K, 0.78 ps at 100 K, and 1.7 ps at 40 K.

With the further decrease in the temperature, max$|\Delta R(t_d)/R|$ gradually decreases, and the rise time of $|\Delta R(t_d)/R|$ becomes slightly faster, while the time characteristics of $\Delta R(t_d)/R$ are qualitatively unchanged even below the critical temperature $T_c = 27$ K, at which the antiferromagnetic transition and dielectric anomaly occur[9]. If the inversion symmetry is lost, the second-order nonlinear optical response such as SHG should be observed. Note that in organic molecular materials such as TTF-CA and $\alpha$-(ET)$_2$I$_3$, in the ferroelectric phases, the reflectivity changes are proportional to the applied terahertz electric fields; moreover, the second-order-nonlinear optical responses were observed[16,39]. However, similar responses, as well as SHG, are not observed in $\kappa$-Cl down to 10 K. This indicates that the inversion symmetry continues to exist at a macroscopic scale below 40 K.

As observed in Figs. 6a, b and 7a, max$|\Delta R(t_d)/R|$ takes the maximum value at approximately 40 K. It is natural to relate the decrease of $\Delta R(t_d)/R$ below 40 K to the change in the dielectric responses. The black solid line in Fig. 7a represents the low-frequency ($\sim$2.1 Hz) dielectric constant previously reported[9]. Its temperature dependence is considerably different from that of max$|\Delta R(t_d)/R|$. The different temperature dependence between the low-frequency dielectric constant and max$|\Delta R(t_d)/R|$ can be explained with the time scale of the measurements. The low-frequency dielectric constant is dominated by slow dynamics such as domain wall motions. It is difficult to derive purely electronic responses occurring in the sub-picosecond time scale from the

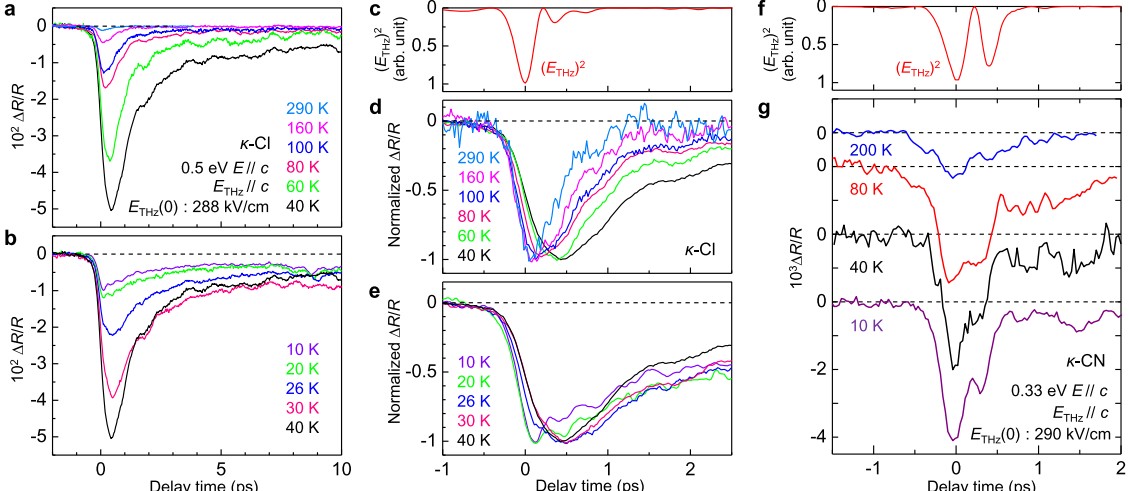

**Fig. 6 Temperature dependence of time evolutions of reflectivity changes in $\kappa$-Cl and $\kappa$-CN. a, b** Time evolutions of $\Delta R(t_d)/R$ at 0.5 eV with $E_{THz}(0) = 288$ kV/cm, $E_{THz}//c$ and $E//c$ at various temperatures in $\kappa$-Cl **a** above 40 K and **b** below 40 K. **c** The square of the terahertz electric-field waveform, $[E_{THz}(t_d)]^2$ in the measurements on $\kappa$-Cl. **d, e** Time evolutions of $\Delta R(t_d)/R$ shown in the normalized scale **d** above 40 K and **e** below 40 K in $\kappa$-Cl. **f** The square of the terahertz electric-field waveform, $[E_{THz}(t_d)]^2$ in the measurements on $\kappa$-CN. **g** Time evolutions of $\Delta R(t_d)/R$ at 0.33 eV with $E_{THz}(0) = 290$ kV/cm, $E_{THz}//c$ and $E//c$ at various temperatures in $\kappa$-CN.

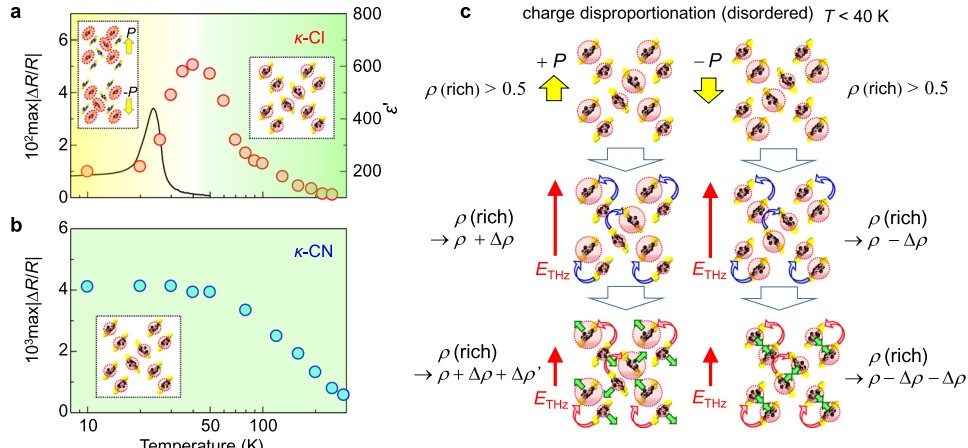

**Fig. 7 Temperature dependence of magnitudes of reflectivity changes in $\kappa$-Cl and $\kappa$-CN. a, b** Temperature dependence of max$|\Delta R(t_d)/R|$ in **a** $\kappa$-Cl and **b** $\kappa$-CN reflecting the electric-field-induced polar states. The solid line in **a** shows the dielectric constant at 2.1 Hz along the $b$-axis, which is adapted by permission from Springer Nature Ltd.: ref. [9] copyright (2012). The insets show a Mott-insulator with homogeneous charge distributions and two kinds of charge-order domains oppositely polarized in steady states. **c** Charge and molecular dynamics induced by the terahertz electric field below 40 K. The initial state consists of two kinds of polar charge-order domains whose polarizations are opposite to each other (the upper part). The figures in the middle and lower parts show schematically charge and molecular dynamics in each domain by a terahertz electric field.

detected signals. On the other hand, the maximum reflectivity change, max$|\Delta R(t_d)/R|$, by the terahertz electric field does not include slow dynamics but includes only the sub-picosecond electronic responses. This is because a terahertz electric field is applied only for 1 ps.

Considering this difference in the two measurement methods, we can explain the observed time evolutions of $\Delta R(t_d)/R$ and their temperature dependence as follows (see also Fig. 7c). Above 40 K, the system is a homogeneous Mott insulator and the applied electric field induces a transient polar CO as illustrated in Fig. 4b, c. With the decrease in temperature below 40 K, the charge disproportionation in each dimer is expected to increase in the steady-state as shown in the upper part of Fig. 7c. A divergent behavior of the dielectric constant around $T_c = 27$ K is attributable to the increase of the dipole moment in each dimer and the

increased ease of its alignment by a quasi-static electric field. Meanwhile, no second-order nonlinear optical signals are observed even at 10 K. This indicates that the charge disproportionation is not completely ordered and that domain structures are likely to be formed in the microscopic scale below $T_c = 27$ K. In the IR molecular vibrational spectroscopy in the steady-state, the charge disproportionation is negligible[32] so that the charge disproportionation in each dimer is likely to be fluctuating in time and space. Here, we assume that the amounts of oppositely polarized CO domains are identical. In this case, the responses of the two kinds of domains to the electric field shown in the middle and bottom parts of Fig. 7c are canceled with each other, and the reflectivity change $\Delta R(t_d)/R$ reflecting the change in the net charge disproportionation decreases as observed. Moreover, the applied electric field might generate a finite

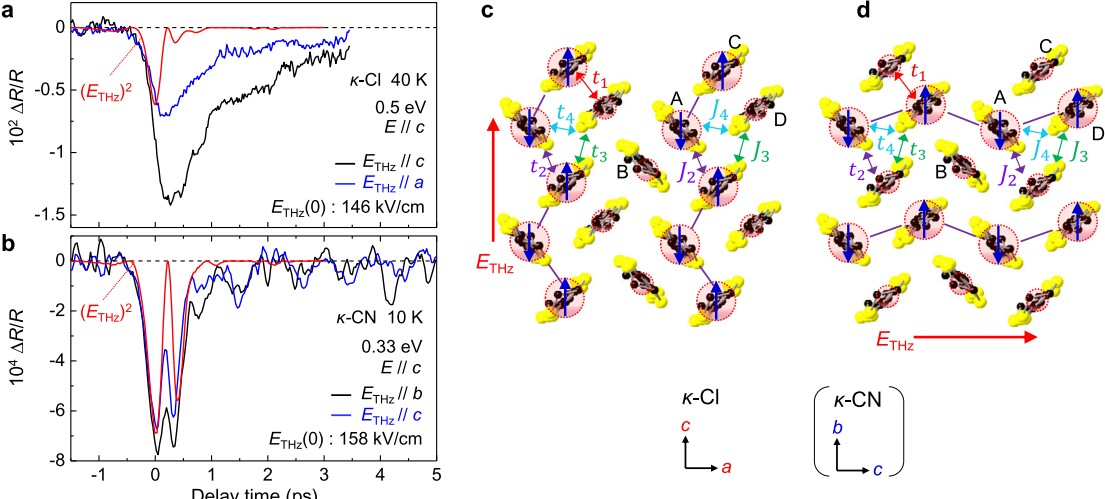

**Fig. 8 Electric-field direction dependence of reflectivity changes and a possible CO state. a** Time evolutions of $\Delta R(t_d)/R$ at 0.5 eV and at 40 K for $E_{THz}//c$ and $E_{THz}//a$ in $\kappa$-Cl. **b** Time evolutions of $\Delta R(t_d)/R$ at 0.33 eV and at 10 K for $E_{THz}//b$ and $E_{THz}//c$ in $\kappa$-CN. Red solid lines in **a** and **b** are the square of the terahertz electric-field waveform $[E_{THz}(t_d)]^2$. **c, d** Two possible CO states. $t_{1-4}$ and $J_{2-4}$ show intermolecular transfer integrals and magnetic interactions, respectively. Red circles and blue arrows represent the charges and spins, respectively. The $c$ and $b$ axis in $\kappa$-CN corresponds to the $a$- and $c$-axis in $\kappa$-Cl, respectively.

polarization and induces SHG. In fact, we observed the electric-field-induced SHG signal, $\Delta I_{SHG}(t_d)$, at 10 K below $T_c = 27$ K, which shows the time evolution different from that of $\Delta R(t_d)/R$ as reported in Supplementary Note 3.

**Transient reflectivity changes by a terahertz electric field in $\kappa$-CN.** We also performed terahertz-pulse-pump optical-reflectivity-probe measurements on $\kappa$-CN. In this compound, the broad intradimer transition band peaked around 0.3 eV exists[40,41]. Therefore, we set the probe pulse at 0.33 eV. Figure 6g shows the time evolutions of $\Delta R(t_d)/R$ at 0.33 eV at several typical temperatures, which are almost the same and roughly proportional to $[E_{THz}(t_d)]^2$ shown in Fig. 6f. $\Delta R(t_d)/R$ also shows a similar time characteristic to $\Delta I_{SHG}(t_d)$ at 50 K (Fig. 2d) reflecting the electric-field-induced polarization, as the case in $\kappa$-Cl (see Supplementary Note 3). Therefore, the time evolutions of $\Delta R(t_d)/R$ should also reflect the charge and molecular dynamics in the electric-field-induced polar state in $\kappa$-CN. Thus, from the results of Fig. 6g, we can consider that no stable CO states are formed by the terahertz electric field down to 10 K in $\kappa$-CN.

Figure 7b shows the temperature dependence of $\max|\Delta R(t_d)/R|$ in $\kappa$-CN. It increases gradually with decreasing temperature; however, it is saturated below 50 K. This behavior is in contrast to that of $\max|\Delta R(t_d)/R|$ in $\kappa$-Cl (Fig. 7a). This result indicates again that the polar CO is difficult to be stabilized in $\kappa$-CN.

**Terahertz electric-field direction dependence of reflectivity changes in $\kappa$-Cl and $\kappa$-CN.** In this section, we compare the dependence of the reflectivity changes $\Delta R(t_d)/R$ on the direction of the terahertz electric field in $\kappa$-Cl and $\kappa$-CN. Figure 8a shows the time evolutions of $\Delta R(t_d)/R$ at 0.5 eV for $E_{THz}//a$ and $E_{THz}//c$ in $\kappa$-Cl. $E_{THz}(0) = 146$ kV/cm, and the electric field $E$ of the probe pulse is parallel to the $c$-axis ($E//c$). The time characteristics for $E_{THz}//a$ and $E_{THz}//c$ are almost identical. The initial decrease in the reflectivity is proportional to $[E_{THz}(t_d)]^2$, represented by the red line; moreover, a finite delayed component appears to exist in common. The maximum value of $\Delta R(t_d)/R$ for $E_{THz}//c$ is two times as large as that for $E_{THz}//a$. This suggests that the charge configuration with the polarization $P//c$ shown in Fig. 8c is more stable than that with $P//a$ shown in Fig. 8d.

Figure 8b shows the time evolutions of $\Delta R(t_d)/R$ at 0.33 eV for $E_{THz}//b$ and $E_{THz}//c$ in $\kappa$-CN with $E_{THz}(0) = 158$ kV/cm and $E//c$. Similarly, as in $\kappa$-Cl, the initial decrease in the reflectivity occurs in accordance with $[E_{THz}(t_d)]^2$ (the red line). At $t_d > 1$ ps, the averaged signal is almost equal to zero, suggesting that no electronic-state changes remain, except for the coherent oscillation attributable to the dimer mode similar to the case in $\kappa$-Cl. This behavior is consistent with the result of the terahertz electric-field-induced SHG. In contrast to the case in $\kappa$-Cl, $\Delta R(t_d)/R$ for $E_{THz}//b$ is equal to that for $E_{THz}//c$. This suggests that the stabilities of two charge configurations with $P//b$ shown in Fig. 8c and with $P//c$ shown in Fig. 8d are not so different from each other in $\kappa$-CN. The reason for such electric-field direction dependences of $\Delta R(t_d)/R$ in $\kappa$-Cl and $\kappa$-CN will be discussed in the Discussion section.

**Discussion**

Here, we discuss the reason for the different electric-field responses between $\kappa$-Cl and $\kappa$-CN; the electric-field-induced polar CO is stabilized in $\kappa$-Cl and not in $\kappa$-CN. In general, a CO is stabilized by inter-site Coulomb interactions. As mentioned in the Result section, in $\kappa$-Cl ($\kappa$-CN), the polarization magnitudes estimated from the terahertz-electric-field-induced SHG signals are approximately 10% (8%) of that ($\sim1$ $\mu$C/cm²) in the ferroelectric CO phase of $\alpha$-(ET)$_2$I$_3$ in the strong electric-field case ($E_{THz}(0) = 288$ kV/cm) (Supplementary Note 1). It makes us expect that the degrees of charge disproportionation induced after the terahertz electric field is applied are comparable to each other and the effects of the inter-site Coulomb interactions would not be so different in the two compounds. The theoretical calculations indicated that the small intradimer transfer integral $t_1$ relative to the interdimer ones $t$ and $t'$ favor CO[14]. However, the average values of $t/t_1$ and $t'/t_1$ in $\kappa$-Cl and $\kappa$-CN are comparable to each other, indicating that $t_1$ does not cause the difference in the two compounds. The theoretical calculations also revealed that the intermolecular Coulomb interaction in $\kappa$-Cl and $\kappa$-CN[28] are almost identical, which cannot be related to the different responses in the two compounds.

Another possible factor stabilizing a CO is an electron-lattice interaction. In the charge-exchange-type CO of the perovskite manganites and the ferroelectric CO of $\alpha$-(ET)$_2$I$_3$, it is known that the electron-lattice interactions play important roles in the

stabilization of the CO states[16,42]. As for the electron-lattice interactions, coherent oscillations observed on the reflectivity changes sometimes include valuable information. In $\kappa$-Cl, the coherent oscillation corresponding to the dimeric molecular displacements is observed on $\Delta R(t_d)/R$ at the intradimer transition as shown in Fig. 5d, which is evidence for the fact that the CO is stabilized via the electron-lattice interaction. In $\kappa$-CN, a similar coherent oscillation attributable to the same origin is observed on $\Delta R(t_d)/R$ at the intradimer transition (Fig. 8b). Judging from these results, the effects of the electron-lattice interactions associated with the dimeric molecular displacements are important in $\kappa$-CN as well as in $\kappa$-Cl.

The other possible origin for the different stability of the polar CO state in the two compounds is the magnetic interaction. In $\kappa$-Cl, the divergent behavior of the low-frequency dielectric response (the black solid line in Fig. 7a) is observed at approximately the antiferromagnetic transition $T_c = 27$ K[9]. This suggests that the polar CO would be related to AFO, while the former does not exhibit a long-range order as mentioned above. It is, however, difficult to demonstrate directly the effects of magnetic interactions on the polar CO. Here, we scrutinize the difference in the intermolecular transfer integrals and resultant intermolecular magnetic interactions in the two compounds and speculate their roles on the stabilization of the polar CO. The values of the intermolecular transfer integrals[28] are shown in Supplementary Note 5.

With regard to the effect of the intermolecular magnetic interactions, when charges in each dimer are disproportionated by the electric field, the antiferromagnetic interactions $J_i \sim \frac{4t_i^2}{U}$ $(i = 2-4)$ determined by the direct intermolecular transfer integrals $t_i (i = 2-4)$ as shown in Fig. 8c might become important rather than the interdimer antiferromagnetic interactions. When the electric field is applied along the $c(a)$-axis in $\kappa$-Cl or along the $b(c)$-axis in $\kappa$-CN, $J_2(J_4)$ stabilizes a spin-singlet in the neighboring two electron-rich molecules, whereas $J_3$ and $J_4(J_2)$ do not. In $\kappa$-Cl, $t_2(J_2) > t_3(J_3) > t_4(J_4)$[13], so that the antiferromagnetic interaction favors the CO polarized along the $c$-axis rather than that along the $a$-axis. This trend is in agreement with the terahertz-electric-field direction dependence of $\Delta R(t_d)/R$ in Fig. 8a. Thus, it is possible to consider that the magnetic interactions $J_2$ play a finite role in the stabilization of the polar CO in $\kappa$-Cl.

The intermolecular magnetic interaction $J$ is estimated to be about 27 meV, when the parameter values of $t_i = 100$ meV and $U = 1.5$ eV are used. Its characteristic time is approximately 150 fs, which is much longer than the characteristic time ($\sim$20 fs) of the intradimer charge transfer determined by $t_1$ ($\sim$200 meV) but shorter than the time scale of the molecular displacements with the period of 1.15 ps. Therefore, the spin relaxations will occur subsequent to the initial charge disproportionation and before the molecular displacements, although it is difficult to discriminate the effect of spin relaxations in the time domain with the time resolution of our measurements ($\sim$100 fs).

Meanwhile, in $\kappa$-CN, $t_3(J_3) > t_2(J_2) > t_4(J_4)$[28] so that the energy gain in the polar CO owing to the local antiferromagnetic interactions is relatively marginal in both $E_{THz}//b$ and $E_{THz}//c$. This trend again corresponds well to the terahertz-electric-field direction dependence of $\Delta R(t_d)/R$ in Fig. 8b, in which the magnitude of $\Delta R(t_d)/R$ are comparable to each other for $E_{THz}//b$ and $E_{THz}//c$, and their decay times are very short in both electric-field directions. This fact and the similar $t$ and $t'$ values hinder the stabilization of the polar CO and AFO in $\kappa$-CN. Thus, the coupling of charge and spin degrees-of-freedom or equivalently the multiferroic nature might be a key factor for stabilizing the polar CO under electric fields. To conclude the roles of magnetic interactions on the polar CO formation, further studies are needed from the viewpoints of both theory and experiment.

Finally, we briefly discuss the nature of the electronic state in $\kappa$-Cl below 40 K. As mentioned in the Result section, we consider that the microscopic polar CO domains are grown below 40 K toward to $T_c = 27$ K. Since no SHG is observed in the steady-state, no net polarization is generated even below $T_c = 27$ K probably due to the disorders or fluctuations of microscopic polar domains. This picture is supported by the comparison of the time evolutions of $\Delta I_{SHG}(t_d)$ and $-\Delta R(t_d)/R$. Above 40 K, the system has no charge disproportionation in the steady-state. In this case, the electric-field-induced polarization reflected by $\Delta I_{SHG}(t_d)$ is created by the charge disproportionation reflected by $-\Delta R(t_d)/R$. Therefore, $\Delta I_{SHG}(t_d)$ and $-\Delta R(t_d)/R$ show almost the same characteristics (see Fig. S1 in Supplementary Note 3). Below 40 K, at which the same amounts of oppositely polarized domains exist, the electric-field-induced changes of the charge disproportionation cancel, resulting in the decrease of $\max|\Delta R(t_d)/R|$ as observed in Fig. 7a, while a finite polarization should be induced by the electric field even below $T_c = 27$ K. In fact, we observed the electric-field-induced SHG signal at 10 K, the time evolution of which is considerably different from that of the reflectivity change $-\Delta R(t_d)/R$ (Supplementary Note 3). This supports our interpretation that the low-temperature phase consists of disordered or fluctuated microscopic CO domains.

In summary, in the present study, we demonstrated in a 2D Mott insulator of an organic molecular compound, $\kappa$-(ET)$_2$Cu[N(CN)$_2$]Cl, that a polar charge order was created via collective intermolecular charge transfers by a strong terahertz electric-field pulse, which is a hidden state never stabilized in the steady-state even at low temperatures. We also ascertained that in an isostructural Mott insulator, $\kappa$-(ET)$_2$Cu$_2$(CN)$_3$, exhibiting a spin-liquid state, a similar polar charge order is not created by the same terahertz pulse excitation. By scrutinizing the intermolecular transfer integrals and magnetic interactions on the 2D molecular planes in addition to the results of terahertz-pump SHG-probe and optical-reflectivity-probe measurements, we suggested an important aspect of $\kappa$-(ET)$_2$Cu[N(CN)$_2$]Cl that coupling of charge and spin degrees of freedom would play an important role in the stabilization of the polar charge order as well as the intermolecular Coulomb interaction and the electron-lattice interaction. As a final remark, our approach using a terahertz electric-field pulse excitation is effective both for impelling an ultrafast switching of electronic phase via dissipation-less electron transfer processes and for understanding the hidden ferroelectric nature of correlated electron materials.

## Methods

**Sample preparations.** Single crystals of $\kappa$-(ET)$_2$Cu[N(CN)$_2$]Cl ($\kappa$-Cl) and $\kappa$-(ET)$_2$Cu$_2$(CN)$_3$ ($\kappa$-CN) were grown by the electrochemical methods[20,21]. In $\kappa$-Cl ($\kappa$-CN), the platelet single crystals having the $ac$ ($bc$) plane as the largest plane were obtained. These planes are parallel to the conducting layer. Typical size of the largest planes of the obtained crystals is 1 mm × 1 mm and a typical crystal thickness is 0.2 mm. All the optical measurements were performed on these largest planes.

**Optical reflection spectroscopy.** For the measurements of the steady-state polarized reflectivity ($R$) spectra in the mid-IR region (0.06–1 eV), a Fourier-transform IR spectrometer equipped with an optical microscope and a wire-grid polarizer was used. The detector was an HgCdTe photovoltaic cell. Samples were cooled in a conduction-type cryostat with a diamond window at the cooling rate of 0.3 K/min. We use the Kramers–Kronig transformation to obtain the optical conductivity ($\sigma$) spectra from the polarized reflectivity spectra.

**Terahertz-pump optical-probe spectroscopy.** In the terahertz-pump optical-probe measurements, a Ti:Sapphire regenerative amplifier (the repetition rate of 1 kHz, the photon energy of 1.55 eV, and the pulse width of 90 fs) was used as a light source. The output from the regenerative amplifier was divided into two beams. One beam was introduced into a nonlinear optical crystal LiNbO$_3$ to generate intense terahertz pulses by the optical rectification with a tilted-pump-pulse-front scheme[17,18]. An electric-field waveform of a terahertz pulse was measured by an electro-optical sampling method. The other beam was used to excite an optical parametric amplifier, from which a probe pulse of 0.06–0.7 eV was obtained.

In the terahertz-pump SHG probe measurements, we used a probe pulse with the photon energy $\hbar\omega = 0.95$ eV and detected its second harmonics ($2\hbar\omega$) with the photon energy of 1.9 eV in the reflection configuration. The diameters of the terahertz pump pulse and the optical probe pulse were 300 and 100 μm, respectively. The electric fields of both pulses were parallel to the $c$-axis. In the terahertz-pump optical-reflectivity-probe measurements, we change the electric fields of the terahertz pulses by using two wire-grid polarizers[39,43].

## Data availability

The data that support the plots within this paper and other findings of this study are available from the corresponding author upon reasonable request.

## Code availability

The codes are available from the corresponding author upon reasonable request.

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

## Acknowledgements

This work was partly supported by Grants-in-Aid for Scientific Research from the Japan Society for the Promotion of Science (JSPS) (Project Numbers JP16H04010, JP17K18746, JP18H01166 and JP18H05225), Nanotechnology Platform Program (Molecule and Material Synthesis) of the Ministry of Education, Culture, Sports, Science and Technology (MEXT), Japan, MEXT Grants-in-Aid for Scientific Research on Innovative Areas "Hydrogenomics" (No. JP18H05516), and CREST, Japan Science and Technology Agency (grant No. JPMJCR1661). H. Yamakawa, T. Morimoto, and T.T. were supported by the JSPS through the Program for Leading Graduate Schools (MERIT). H. Yamakawa and T. Morimoto were supported by the JSPS Fellowship Research Program for Young Scientist.

## Author contributions

K.M. and K.K. provided single crystals of $\kappa$-Cl and $\kappa$-CN. M.S. and H.M.Y. provided thin single crystals of $\kappa$-Cl. H.M. provided single crystals of $\alpha$-(ET)$_2$I$_3$. H. Yamakawa measured the steady-state reflectance spectra. H. Yamakawa, T. Miyamoto, T. Morimoto, T.T., and N.K. constructed the terahertz-pump optical-probe systems. H. Yamakawa, N.T., S.L., and H. Yoshimochi performed the measurements. H.O. coordinated the study. H. Yamakawa and H.O. wrote the paper with inputs from all authors.

## Competing interests

The authors declare no competing interests.
