## [Peer Review File · Nature Communications]

REVIEWER COMMENTS

Reviewer #1 (Remarks to the Author):

The manuscript reports ultrafast optical measurements of two organic Mott insulating compounds. As far as I am aware, this is a novel set of data. It should likely be reported somewhere in a different format. I am, however, sorry to say that the manuscript is missing several crucial elements such as a clear explanation of how the data supports the conclusions, a reasonable attempt to set the current work in the proper context of what has been done previously, and a clear delineation of when the authors are reviewing prior work and making new insights based on their work. These elements need to be clarified before the work can reasonably be reviewed for Nature Communications. I think the journal should not consider the manuscript further.

1. The introduction provided does not make a reasonable attempt to put the current study in the context of the literature. If the key selling point is "THz induced ferroelectricity", the introduction must put the current study in the context of prior works on this subject such as [Science 364, 1079–1082 (2019), Science 364, 1075–1079 (2019)] which report exactly that. The former reference is included in a way that doesn't clarify the contents and context of the paper and the latter appears to have been missed entirely.

2. While I agree that the second harmonic generation signal *could* be used to prove the existence of a ferroelectric state, the current work neglects its duty to provide a clear and direct explanation of the connection between the experimental data and the assertion.

3. The discussion/interpretation section rests very heavily on prior work. It was very difficult to find a physical insight that could not have been made on the basis of prior work. The manuscript cannot reasonably be reviewed for Nat. Comm. unless it makes its case for unique new insights.

Reviewer #2 (Remarks to the Author):

The reviewer apology for the delayed response.

This paper by Yamakawa, and co-workers is focused on addressing an interesting but highly challenging problem, namely the possibility to create order, and therefore functions, by light. More precisely, the authors used a THz pulse to induce transient polar order in κ -(ET)₂Cu[N(CN)₂]Cl. These types of compounds have garnered considerable interest over the years due to their Mott-Insulator nature and different instabilities related to charge order, charge disproportionation or electronic ferroelectricity.

What the present contribution seeks to do is apply ultrafast time-resolved THz pump and optical reflectivity or SHG probe techniques to track stabilization of electronic ferroelectricity and deconvolve the structural and electronic dynamics.

The paper and supplementary information present a large number of very interesting data and analysis, of potential interest for Nature communication. Many data are carefully presented in figure and tables and the data analysis is quite robust. The authors claim to provide evidence about the formation of polar charge order (CO) and bring information about the important factors stabilizing the polar CO state. A clear threshold response to the THz excitation field is reported. The supplementary

section is well documented and presented, which helps understanding the paper. However, the discussion part and interpretation of the data looks half-baked. I think some data and important discussions are missing for the interpretation.

For this reason, I recommend not to publish the paper in its present form. The paper may be reconsidered once the authors will have taken into account the following comments and questions.

- 1 The main problem in the presentation of the paper is that a conclusion or summary part is missing for underlying the main results, findings or interpretation, just before the method section.
- 2 The photoinduced polar CO is a very interesting topic. Fig. 3 and 4 show that after 10 ps the systems remains in a transient state. Is there SHG data showing how long the polar state survives for K-Cl above threshold? What is the lifetime of the polar order exactly? Does it depends on temperature?
- 3 Due to the molecular packing, the dimers are tilted with respect to the c axis. A THz field may induce intra-dimer CT, resulting in a CT dipole for the dimer. Each dimer has a dipole component along c or perpendicular to the c axis. The THz pulse may be polarized // c or perpendicular to c axis as well. In the first case, the charge order results in a total polarization // c as shown in the middle panel of Fig. 1b. All the dipoles point up along c for example, with perpendicular components opposite from one dimer to the adjacent one, resulting in a net polarization along c only by symmetry. However, when the THz pulse is polarized perpendicular to the c axis, the charge order results in a total polarization perpendicular to c . The intra-dimer dipole are the same, but within the crystal it is now the perpendicular component which add in a constructive way with a total net polarization perpendicular to c , while the components of the dipoles cancel along c from one dimer to another one. The CO shown in the midel panel of fig. 1b correspond to the one for a THz polarization along c and the CO mentioned above for the THz polarization perpendicular to c must also be shown and discussed.
- 4 This brings my first question when authors compare the response to THz excitation parallel or perpendicular to c . The author mention "The maximum value of $\Delta R(td)/R$ for $E_{THz} \parallel c$ is two times as large as that for $E_{THz} \parallel a$. This suggests that the charge configuration with the polarization $P \parallel c$ is more stable than that with $P \parallel a$ ". I don't understand how the observation brings to this conclusion. If we consider the intradimer dipole, we may split the components: $P = P \cos \alpha + P \sin \alpha$ along the c axis or perpendicular. The angle α with respect to c seems to be in the 30° range. Since the intensity of the SHG changes as P^2 and depends on the angle α between the dipoles and the THz field, and $E_{THz} \cos \alpha$, and it should be the same for ΔR . Then, it seems natural with $\alpha = 30^\circ$ that the maximum value of $\Delta R(td)/R$ for $E_{THz} \parallel c$ is two times as large as that for $E_{THz} \parallel a$ ". The authors should discuss the possibility mentioned above that THz polarization parallel or perpendicular to the c axis may generate to types of CO with net polarization along c or perpendicular to c , as required by symmetry.
- 5 There is a similar story with the vibration observed. THz polarization parallel or perpendicular to c may induce molecular motions with a net polarization due to CT dipoles and molecular motions parallel or perpendicular to c . When THz pump and the probe are set // c the probe is sensitive to the components of the dimerization modes of the different dimers, whose component along c breathe in phase along c . This explains why oscillations are observed for the THz pump and THz probe parallel to c . However, when the THz pump is perpendicular to c and the probe is set // c , the probe is still sensitive the component along c of the breathing dimers, and these components are out of phase now from one dimer to the other one. This should explain why no oscillations are observed when the THz pump is perpendicular to c and the probe are set // c . It is then necessary to show data for a THz pump set perpendicular to c and a probe also set perpendicular to c . This may reveal another mode related to a different charge order. The two modes (the one observed and the one which I propose to track) are two IR-active modes with different polarization.

- 6 One important difference between K-CN and K-Cl are the anions. I am also wondering if the anions can play a role in the stabilization of the polar order. Indeed, anions must also move under the effect of the THz pulse. Is there any relevant anion packing/motions that may help stabilizing the polar CO in K-Cl and not in K-CN? This point should be discussed as it may seem to be an obvious explanation to the general reader of the different response of both compounds.

- 7 In Fig. 1e, is there any signature of vibration modes or oscillation (FFT)?

- 8 When the authors write "The decrease in the dimerization and the resultant decrease in t_1 in each dimer should also induce additional charge transfers and increases the charge disproportionation further.", can author provide an estimate of the motions?

- 9 The analysis of the spectral change is misleading in the way it is presented. The authors analyse the data change "By assuming a blue shift (6.1 meV) of the interdimer transition, and a small blue shift (1.8 meV) and an intensity decrease (2.9%) of the intradimer transition, we can approximately reproduce the $\Delta R(td = 0 \text{ ps})$ spectrum,". I suppose they analysed the data, interpreted the spectral changes in terms of shifts of band and intensities and then refined the spectral shifts and intensity changes. The data interpretation and analysis should be presented in a more scientific way, not "assuming" something but explaining why the data change should be interpreted in this way.

Other minor points:

- p15 the authors mention "SHG are not observed in κ -Cl down to 10 K. This indicates that the inversion symmetry continues to exist at a macroscopic scale below 40 K." Did the authors track SHG for light polarization // c only, or did they try perpendicular to c as well? Indeed polar order could appear in different directions depending on the nature of the symmetry breaking.

- p6 it is written " Δ /SHG(td) increases in accord to $[E_{\text{THz}}(td)]^2$ (the red line in the middle panel) without delay, indicating that the initial response is electronic in nature". However, there are some intra-dimer/molecular modes, like molecular bending, which may carry a dipole. Such intra-molecular bending mode may fall in the 100 cm^{-1} range, with $\frac{1}{4}$ period shorter than 75 fs. Therefore I don't understand why the contribution of molecular motion can be ruled out.

- The figures are small and hard to read. Especially the figures showing the molecular structures and packing (1b, 2h, 5c). It would be much easier to read with molecules (or dimers) represented by strings (or pairs of strings) like in Gati's PRL for example (120, 247601, 2018). The molecular view with atoms too small to be distinguished is useless. Or a larger figure must be used (for the first one may be). It will be much easier to read the CO as well, without atomic balls superposing to CO balls.

- It is mentioned in the text p 5 "we set the photon energy of the incident pulse to be 0.95 eV, since κ -Cl and κ -CN are almost transparent for this pulse". However, in sup. Info. It is written that the absorption depth is about 1 μm

- in equation (1), for the fit of the data, the oscillating part is written with a "c" subscript, while in the text it is an "osc" subscript. Please check consistency. In the text τ_d is also mentioned and it should be more clearly defined.

- p12 "The parameter values adopted are..." error bars should be given for the values extracted from the fit. A table listing the results of the fits in sup info would help also.

- p14 when authors mention " the potential barrier seems to be very small". What is the order of magnitude expected compared to thermal or electric energy?

The writing and language should be improved. Some sentences are not fluent, hard to read due to wording or typos, like

"much attention not only from as a new topic in solid state physics"

"rapidly decreases to zero at when the electric field crosses zero"

" As the electric-field amplitude increases, both of the initial decrease of the reflectivity and its recovery become slow." -> slows down?

Parallel should be written "// "not "||"

In sup info $\Delta P_{2\kappa-\text{Cl}}$ should be written $(\Delta P_{\kappa-\text{Cl}})^2$

I hope the authors will be able to resubmit an improved version of the paper, as this field of THz pump order is of great potential interest.

Reviewer #3 (Remarks to the Author):

The ultra-fast optical of field-induced stimulation of phase transitions in complex materials is a topic which is currently very actively investigated. In particular, the possibility of inducing orders, rather than melting it, is a central goal of this field. To my knowledge, there are no examples in which this has been unambiguously demonstrated and in which the underlying mechanism is entirely understood. The present study provides a possible candidate for a field-induced order which may be driven to large extent by electronic correlations. It reports a long-lived response of the organic Mott insulator $\kappa\text{-Cl}$ to strong THz fields (with a strong nonlinear dependence of the relevant signal on the THz field), while no corresponding signal is observed in a related compound ($\kappa\text{-CN}$). The results are based on two complementary measurements [second harmonic generation and reflectivity measurements], and the clearly presented analysis thoroughly covers relevant questions, in particular the temperature and field dependence of the signal. The dependence on the field amplitude shows a clear nonlinearity, which supports a field-induced non-equilibrium transition. I therefore believe that the paper is scientifically sound, clearly presented, its topic should be of broad interest and definitely stimulate further theoretical and experimental work.

I have some questions concerning the analysis of the temperature dependence: The analysis in Fig. 5 is based on the maximum reflectivity change $\Delta R/R$. The later seems to be mostly dominated by process "alpha" in Fig.4, i.e., the instantaneous response to the THz pulse. On the other hand, Fig 4f shows that process "alpha" is proportional to the square of the THz field down to low THz amplitudes, and therefore should relate to the equilibrium properties of the material rather than a non-equilibrium phase.

This would indicate that the 40K temperature scale has a meaning already for the equilibrium properties. Is this correct? Later the authors discuss a picture in which the low T phase may be interpreted as ordered domains. Would that mean that the 40K scale is related to the onset of the domain order, different from the 27K phase transition?

The beta process, on the other hand, has a clearly nonlinear dependence of the field, with different regimes at large and small fields. To characterize the temperature-dependence of the non-equilibrium transition, would it be possible to provide both an analysis of the temperature dependence of the alpha process and the "beta" process?

However, apart from these minor comments, I find this a stimulating work, and I support publication in Nature Communications if the authors can comment on the above issues.

Replies to comments of Reviewer #1

We thank Reviewer #1 for his/her careful reading of our paper and valuable comments. We divided the comments of Reviewer #1 to three parts (1)-(3). We show the reply to each comment below.

Comment (1) of Reviewer #1

The introduction provided does not make a reasonable attempt to put the current study in the context of the literature. If the key selling point is “THz induced ferroelectricity”, the introduction must put the current study in the context of prior works on this subject such as [*Science* **364**, 1079–1082 (2019), *Science* **364**, 1075–1079 (2019)] which report exactly that. The former reference is included in a way that doesn’t clarify the contents and context of the paper and the latter appears to have been missed entirely.

Reply to comment (1) of Reviewer #1

Taking the comment of Reviewer #1 into account, we revised the introduction in which we quoted two references raised by Reviewer #1 as follows.

[Line 8-13 in page 3 of the new MS]

“In fact, it was reported that a macroscopic polarization was generated by an electric-field component of a terahertz pulse in an organic molecular compound, tetrathiafulvalene-*p*-chloranil, in the paraelectric phase⁵ and a transition metal oxide SrTiO₃ in the quantum paraelectric phase⁶. In addition, a polarization control of SrTiO₃ was successfully made by exciting a specific lattice vibration with a mid-infrared pulse⁷.”

We added Ref. 7 in the reference section as follows.

“7. Nova, T. F., Disa, A. S., Fechner, M. & Cavalleri, A. Metastable ferroelectricity in optically strained SrTiO₃. *Science* **364**, 1075-1079 (2019).”

We also changed the reference numbers appropriately.

Comment (2) of Reviewer #1

While I agree that the second harmonic generation signal *could* be used to prove the existence of a ferroelectric state, the current work neglects its duty to provide a clear and direct explanation of the connection between the experimental data and the assertion.

Reply to comment (2) of Reviewer #1

As the referee commented on, in our study we used the second harmonic generation (SHG) as the evidence of a ferroelectric state. The detailed explanations about the connection between the experimental data and the quantitative evaluation of the macroscopic polarization induced by the

terahertz electric field are reported in Supplementary Note 1. To make clear the content of Supplementary Note 1, we modified the title of this Note as follows.

[Line 1 in page 2 of the old Supplementary Information]

S1. Evaluation of terahertz-field-induced polarization magnitudes in κ -(ET)₂Cu[N(CN)₂]Cl

[Line 1 in page 2 of the new Supplementary Information]

S1. Evaluation of polarization magnitudes from terahertz-pump SHG-probe measurements in κ -(ET)₂Cu[N(CN)₂]Cl

We also report the comparison of the electric-field-induced SHG and reflectivity changes in Supplementary Note 3 in the Supplementary Information. The results and discussions in this Note support our interpretation that the observed SHG signals reflect the ferroelectric polarization appearing in the electric-field-induced charge order.

Comment (3) of Reviewer #1

The discussion/interpretation section rests very heavily on prior work. It was very difficult to find a physical insight that could not have been made on the basis of prior work. The manuscript cannot reasonably be reviewed for Nat. Comm. unless it makes its case for unique new insights.

Reply to comment (3) of Reviewer #1

We thank Reviewer #1 for his/her valuable comment. Taking this comment into account, we corrected the introduction to make clear the main purpose and the most important achievement of our study as follows.

[Line 14-15 in page 3 of the new MS]

“Here, we focus on the creation of macroscopically polar state by an irradiation of a nearly monocyclic terahertz pulse via a new kind of electric-field-induced phase transition.”

[Line 11-17 in page 5 of the new MS]

“In this paper, we report that a macroscopically polar state is generated by a terahertz electric field in κ -Cl via a novel mechanism, that is the electric-field-induced Mott-insulator to CO transition. From the comparative studies of terahertz-pump optical-probe spectroscopy on κ -Cl and κ -CN, we clarify the roles of the intermolecular Coulomb interaction, the electron-lattice interaction, and the magnetic interaction on the stabilization of the polar CO in κ -Cl under strong electric fields. The findings obtained in this study provide new physical insights on the Mott physics in organic molecular materials.”

In addition, we added the summary paragraph at the end of the new MS, in which we summarized the achievements and physical insights obtained in our study as follows.

[Line 15 in page 23- line 7 in page 24 of the new MS]

“In summary, in the present study, we demonstrated in a two-dimensional Mott insulator of an organic molecular compound, κ -(ET)₂Cu[N(CN)₂]Cl, that a polar charge order state was created via collective intermolecular charge transfers by a strong terahertz electric-field pulse. We also ascertained that in an isostructural Mott insulator, κ -(ET)₂Cu₂(CN)₃, exhibiting a spin-liquid state at low temperatures, a similar polar charge order is not stabilized by the same terahertz pulse excitation. By scrutinizing the intermolecular transfer energies and magnetic interactions on the two-dimensional molecular planes in addition to the results of terahertz-pump second-harmonic-generation-probe and optical-reflectivity-probe measurements, we suggested an important aspect of κ -(ET)₂Cu[N(CN)₂]Cl that a coupling of charge and spin degrees of freedom would play an important role in the stabilization of the electric-field-induced polar charge order as well as the intermolecular Coulomb interaction and the electron-lattice interaction. As a final remark, our approach using a terahertz electric-field pulse excitation is effective both for impelling an ultrafast switching of electronic phase via dissipation-less electron transfer processes and for understanding the hidden ferroelectric nature of correlated electron materials.

”

Replies to comments of Reviewer #2

We thank Reviewer #2 for his/her careful reading of our paper and valuable comments. We thank Reviewer #2 for recognizing the value of our paper and saying “The paper and supplementary information present a large number of very interesting data and analysis, of potential interest for Nature communication.”

We divided the comments of Reviewer #2 to seventeen parts (1)-(17). We show the reply to each comment below.

Comment (1) of Reviewer #2

The main problem in the presentation of the paper is that a conclusion or summary part is missing for underlying the main results, findings or interpretation, just before the method section.

Reply to comment (1) of Reviewer #2

We thank Reviewer #2 for his/her valuable comment. In a standard style of Nature Communications, a conclusion section is not set. Taking the comment into account, we added the paragraph of the summary at the end of the new MS as follows.

[Line 15 in page 23 to line 7 in page 24 of the new MS]

“In summary, in the present study, we demonstrated in a two-dimensional Mott insulator of an organic molecular compound, κ -(ET)₂Cu[N(CN)₂]Cl, that a polar charge order state was created via collective intermolecular charge transfers by a strong terahertz electric-field pulse. We also ascertained that in an isostructural Mott insulator, κ -(ET)₂Cu₂(CN)₃, exhibiting a spin-liquid state at low temperatures, a similar polar charge order is not stabilized by the same terahertz pulse excitation. By scrutinizing the intermolecular transfer energies and magnetic interactions on the two-dimensional molecular planes in addition to the results of terahertz-pump second-harmonic-generation-probe and optical-reflectivity-probe measurements, we suggested an important aspect of κ -(ET)₂Cu[N(CN)₂]Cl that a coupling of charge and spin degrees of freedom would play an important role in the stabilization of the electric-field-induced polar charge order as well as the intermolecular Coulomb interaction and the electron-lattice interaction. As a final remark, our approach using a terahertz electric-field pulse excitation is effective both for impelling an ultrafast switching of electronic phase via dissipation-less electron transfer processes and for understanding the hidden ferroelectric nature of correlated electron materials.”

Comment (2) of Reviewer #2

The photoinduced polar CO is a very interesting topic. Fig. 3 and 4 show that after 10 ps the systems remains in a transient state. Is there SHG data showing how long the polar state survives for K-Cl above threshold? What is the lifetime of the polar order exactly? Does it depends on temperature?

Reply to comment (2) of Reviewer #2

In the SHG measurements, we cannot use the transmission configuration since both of the probe light and the second harmonic light are absorbed in the crystal. Therefore, we adopted the reflection configuration, in which the SHG signals become very small as compared to those in the transmission configuration. It is because the coherence length of SHG in the reflection configuration is much shorter than that in the transmission configuration. The details of the experimental conditions of SHG are reported in Supplementary Note 1. As shown in Supplementary Figure 1, at 40 K the time characteristic of the SHG signal and that of the reflectivity change at 0.5 eV are almost the same with each other. Based upon this fact, we consider that the dynamics of the reflectivity change reflects the dynamics of the electric-field-induced polar CO. In this study, therefore, the reflectivity change is used as a probe to investigate the detailed behaviors of the electric field-induced polar CO.

The decay of the reflectivity changes at 0.5 eV is expressed by the first term and the second term in the function $F(t)$ of eq. (1). The first term shows an electronic process, that is, the intradimer charge transfer, which is called process α . Its time constant τ_1 is 0.3 ps. The second term reflects the nonlinear transition with the structural change, that is, the transition to the metastable polar CO, which is called process β and is phenomenologically expressed by two exponential functions with time constants of 0.92 ps and 27 ps. In the case that the electric-field amplitude is larger than 200 kV/cm at 40K, process β dominates the signal magnitude (see Fig. 3d and Fig. 3f). In order to see how the lifetime of the polar CO state depends on the electric-field amplitude, the characteristic time τ_d corresponding to the lifetime is plotted as a function of the electric-field amplitude in Fig. 3g. (As for the definition of τ_d , see the reply to the comment (14) of Reviewer #2.) At 288 kV/cm, it is about 2 ps. Detailed analysis using eq. (1) at different temperatures were not performed. However, by using the value of τ_d as a measure of the life time of the polar CO state, we can see that the lifetime increases with decrease of temperature down to 40 K; $\tau_d \sim 0.44$ ps at 290 K, 0.78 ps at 100 K, and 1.7 ps at 40 K.

In the new MS, we modified the related description as follows.

[Line 9-20 in page 15 of the old MS]

“With the decrease in the temperature from 290 K, $\max|\Delta R(t_d)/R|$ increases; it attains the maximum value at 40 K. Simultaneously, the rise time of $|\Delta R(t_d)/R|$ increases and the decay time of $|\Delta R(t_d)/R|$ is prolonged as seen in Fig. 4d. This suggests that the β component reflecting the stabilization process of the polar CO state is enhanced with the decrease in the temperature. In fact, the time characteristic of $\Delta R(t_d)/R$ at 290 K shown in Fig. 4d is very similar to that of the α component of $\Delta R(t_d)/R$ at 40 K shown in Fig. 3d and the β component does not exist at 290 K. All these results indicate that at approximately 40 K, the instability to the polar CO state is maximized and the electric-field-induced CO is most likely to be stabilized. By using the value of τ_d obtained from the decay dynamics of $\Delta R(t_d)/R$ $|\Delta R(t_d)/R|$, we can see that the life time of the polar CO state increases with decrease of temperature down to 40 K, e.g., $\tau_d \sim 0.44$ ps at 290 K, 0.78 ps at 100 K, and 1.7 ps at 40 K.”

Comment (3) of Reviewer #2

Due to the molecular packing, the dimers are tilted with respect to the c axis. A THz field may induce intra-dimer CT, resulting in a CT dipole for the dimer. Each dimer has a dipole component along c or perpendicular to the c axis. The THz pulse may be polarized // c or perpendicular to c axis as well. In the first case, the charge order results in a total polarization // c as shown in the middle panel of Fig. 1b. All the dipoles point up along c for example, with perpendicular components opposite from one dimer to the adjacent one, resulting in a net polarization along c only by symmetry. However, when the THz pulse is polarized perpendicular to the c axis, the charge order results in a total polarization perpendicular to c . The intra-dimer dipoles are the same, but within the crystal it is now the perpendicular component which add in a constructive way with a total net polarization perpendicular to c , while the components of the dipoles cancel along c from one dimer to another one. The CO shown in the middle panel of Fig. 1b correspond to the one for a THz polarization along c and the CO mentioned above for the THz polarization perpendicular to c must also be shown and discussed.

Reply to comment (3) of Reviewer #2

In Fig. 6, we show the comparison of the terahertz polarization dependence of the reflectivity changes in κ -Cl (Fig. 6a) and in κ -CN (Fig. 6b). The results are discussed in the section titled "Terahertz electric-field direction dependence of reflectivity changes in κ -Cl and κ -CN" from page 18 to 19. Taking the comment of Reviewer #2 into account, we added a schematic figure of a possible CO state in Fig. 6d when the electric field of the terahertz pulse perpendicular to the c (b) axis in κ -Cl (κ -CN). In the new MS, we also added the explanations of these figures.

Comment (4) of Reviewer #2

This brings my first question when authors compare the response to THz excitation parallel or perpendicular to c . The authors mention "The maximum value of $\Delta R(td)/R$ for $E_{THz} // c$ is two times as large as that for $E_{THz} // a$. This suggests that the charge configuration with the polarization $P // c$ is more stable than that with $P // a$ ". I don't understand how the observation brings to this conclusion. If we consider the intradimer dipole, we may split the components: $P = P \cos \alpha + P \sin \alpha$ along the c axis or perpendicular. The angle α with respect to c seems to be in the 30° range. Since the intensity of the SHG changes as P^2 and depends on the angle α between the dipoles and the THz field, and $E_{THz} \cos \alpha$, and it should be the same for ΔR . Then, it seems natural with $\alpha = 30^\circ$ that the maximum value of $\Delta R(td)/R$ for $E_{THz} // c$ is two times as large as that for $E_{THz} // a$ ". The authors should discuss the possibility mentioned above that THz polarization parallel or perpendicular to the c axis may generate to types of CO with net polarization along c or perpendicular to c , as required by symmetry.

Reply to comment (4) of Reviewer #2

We would like to thank Reviewer #2 for his/her important comment. Here, for simplicity we consider that the material is composed of two kinds of isolated dimers having different directions. The direction of the vector connecting two molecules of each dimer is inclined from the c axis by $+\theta_A$ and $-\theta_B$ ($\theta_A > 0$ and $\theta_B > 0$). The dipole moment μ induced by a certain charge disproportionation in each dimer is assumed to have the same direction as each vector. In this case, The component parallel c , $\mu_{\parallel c}$, and the component perpendicular, $\mu_{\perp c}$, of the sum of two dipole moments induced by the electric field E_{THz} follow the relations below;

For $E_{\text{THz}}//c$,

$$\mu_{\parallel c} \propto [(E_{\text{THz}}\cos\theta_A)(\mu\cos\theta_A) + (E_{\text{THz}}\cos\theta_B)(\mu\cos\theta_B)] = E_{\text{THz}}\mu(\cos^2\theta_A + \cos^2\theta_B)$$

$$\mu_{\perp c} \propto E_{\text{THz}}\mu|\cos\theta_A\sin\theta_A - \cos\theta_B\sin\theta_B|.$$

For $E_{\text{THz}} \perp c$,

$$\mu_{\parallel c} \propto E_{\text{THz}}\mu|\sin\theta_A\cos\theta_A - \sin\theta_B\cos\theta_B|$$

$$\mu_{\perp c} \propto E_{\text{THz}}\mu(\sin^2\theta_A + \sin^2\theta_B).$$

It is natural to consider that the reflectivity change $\Delta R/R$ reflecting the charge disproportionation or charge order is proportional to the sum of the square of the dipole moment in each dimer. Thus, we obtain the following relations;

For $E_{\text{THz}}//c$,

$$\frac{\Delta R}{R} \propto [(E_{\text{THz}}\cos\theta_A)^2\mu^2 + (E_{\text{THz}}\cos\theta_B)^2\mu^2] = (E_{\text{THz}})^2\mu^2(\cos^2\theta_A + \cos^2\theta_B).$$

For $E_{\text{THz}} \perp c$,

$$\frac{\Delta R}{R} \propto [(E_{\text{THz}}\sin\theta_A)^2\mu^2 + (E_{\text{THz}}\sin\theta_B)^2\mu^2] = (E_{\text{THz}})^2\mu^2(\sin^2\theta_A + \sin^2\theta_B).$$

Note that the magnitude of the reflectivity change $\Delta R/R$ for $E//c$ in our case reflects the degree of the charge disproportionation or equivalently the magnitude of CO and does not depend on the direction of the polarization of CO. In the case that $\theta_A + \theta_B \sim 90^\circ$, $\Delta R/R$ values for $E_{\text{THz}}//c$ and $E_{\text{THz}} \perp c$ are also comparable to each other. We consider that this is the case for κ -CN as seen in Fig. 6b. In contrast to the case of κ -CN, in κ -Cl, $\Delta R/R$ shows a strong dependence on the direction of the terahertz electric field; $\Delta R/R$ for $E_{\text{THz}}//c$ is twice as large as that for $E_{\text{THz}} \perp c$. The values of θ_A and θ_B are not so different in the two compounds, so we suppose that in κ -Cl the charge order with the $P//c$ generated for $E_{\text{THz}}//c$ is more stable than that with $P//a$ generated for $E_{\text{THz}} \perp c$ ($E_{\text{THz}}//a$). These discussions are somewhat complicated and not so important for general readers, so that we did not add them in the new MS.

By scrutinizing the molecular arrangements in κ -Cl, we suggest that anisotropic interdimer interactions and intermolecular interactions between two molecules belonging to different dimers would be responsible for the different stability of the charge order with $P//c$ and $P//a$. The detailed discussions are given in the Discussion section from line 15 in page 19 to line 19 in page 22.

Comment (5) of Reviewer #2

There is a similar story with the vibration observed. THz polarization parallel or perpendicular to c may induce molecular motions with a net polarization due to CT dipoles and molecular motions parallel or perpendicular to c . When THz pump and the probe are set $//c$ the probe is sensitive to the components of the dimerization modes of the different dimers, whose component along c breathe in phase along c . This explains why oscillations are observed for the THz pump and THz probe parallel to c . However, when the THz pump is perpendicular to c and the probe is set $//c$, the probe is still sensitive the component along c of the breathing dimers, and these components are out of phase now from one dimer to the other one. This should explain why no oscillations are observed when the THz pump is perpendicular to c and the probe are set $//c$. It is then necessary to show data for a THz pump set perpendicular to c and a probe also set perpendicular to c . This may reveal another mode related to a different charge order. The two modes (the one observed and the one which I propose to track) are two IR-active modes with different polarization.

Reply to comment (5) of Reviewer #2

We would like to thank Reviewer #2 for his/her valuable comment. We did not perform the pump probe measurements with the combination of $E_{\text{THz}}//a$ and $E//a$ in $\kappa\text{-Cl}$. As shown in Fig. 6a and discussed in the section titled “Terahertz electric-field direction dependence of reflectivity changes in $\kappa\text{-Cl}$ and $\kappa\text{-CN}$ ”, the stabilities of the CO states polarized along c and along a are different from each other in $\kappa\text{-Cl}$. In this case, the molecular displacements stabilizing two kinds of CO states might also be slightly different. The vibrations corresponding to those molecular displacements are fundamentally the dimerization modes, which are not infrared active but Raman active in the uniformly charged state. By scrutinizing the coherent oscillations on the reflectivity changes $\Delta R/R$ and the steady-state polarized Raman spectra, we might obtain important information about the difference in the modes stabilizing two CO states. We consider that this is beyond the scope of our study and an important future work.

Comment (6) of Reviewer #2

One important difference between $\kappa\text{-CN}$ and $\kappa\text{-Cl}$ are the anions. I am also wondering if the anions can play a role in the stabilization of the polar order. Indeed, anions must also move under the effect of the THz pulse. Is there any relevant anion packing/motions that may help stabilizing the polar CO in $\kappa\text{-Cl}$ and not in $\kappa\text{-CN}$? This point should be discussed as it may seem to be an obvious explanation to the general reader of the different response of both compounds.

Reply to comment (6) of Reviewer #2

When a terahertz electric field generates a polar CO state, anion displacements might occur as the Reviewer #2 commented on, and they might make a polar CO state stabilized. In our study, from the

analysis of the time characteristics of the reflectivity changes, the initial dynamics of the formation of the polar CO state consists of two processes; the instantaneous response attributed to the intradimer CT and the delayed response with the time constant of 0.28 ps attributed to the molecular displacements corresponding to the release of the dimerization. Other additional responses are not clearly observed. After we received the reviewers' comments, we have found a neutron scattering study on κ -Cl suggesting that phonon frequencies related to anions is low [M. Matsuura *et al.*, Phys. Rev. Lett. **123**, 027601 (2019)]. The frequency of the typical phonon mode is 1.5 meV and its period is 2.8 ps. Therefore, we suppose that the anion dynamics would not play important roles on the initial rapid dynamics of the generation of the polar CO state within 0.3 ps. But this is a speculation and it is difficult for us to discuss the precise roles of anions on the transient polar CO state induced by the terahertz field in the sub-picosecond time domain. To ascertain our speculation, further studies including calculations of anion dynamics and detailed phonon spectroscopy in the very low frequency region below ~ 1 THz (~ 30 cm⁻¹) should be necessary.

Taking these discussions into account, we quoted in the introduction the above-mentioned paper as the study suggesting the importance of the electron phonon interaction. This paper is cited as ref. 15 in the new MS.

[15] M. Matsuura *et al.*, Phys. Rev. Lett. **123**, 027601 (2019).

Comment (7) of Reviewer #2

In Fig. 1e, is there any signature of vibration modes or oscillation (FFT)?

Reply to comment (7) of Reviewer #2

As mentioned in the reply to comment (2) of Reviewer #2, the electric-field induced SHG measurement cannot be performed in the transmission configuration since both of the probe light and the second harmonic light are absorbed in the crystal. Therefore, this measurement is performed in the reflection configuration. In this case, the coherence length of the SHG is very short, so that the SH light is extremely weak and is difficult to measure with a good signal to noise ratio. Therefore, it is also difficult to measure the time characteristic of SHG in a long temporal region. As a result, we cannot extract the vibration from that. It is natural to consider that the same vibration as that observed in the time characteristics of the reflectivity changes is included in the SHG signal, although it is difficult to confirm it in the present study.

Comment (8) of Reviewer #2

When the authors write "The decrease in the dimerization and the resultant decrease in t_1 in each dimer should also induce additional charge transfers and increases the charge disproportionation further.", can author provide an estimate of the motions?

Reply to comment (8) of Reviewer #2

We thank Reviewer #2 for his/her valuable question. The analyses of the time characteristics of the reflectivity changes revealed that the dimerization decreases just after the initial charge disproportionation is induced. It is however difficult to estimate the magnitudes of the molecular displacements. The first principle calculation about the atomic positions and electronic states under an electric field might give their quantitative estimation.

Comment (9) of Reviewer #2

The analysis of the spectral change is misleading in the way it is presented. The authors analyse the data change " By assuming a blue shift (6.1 meV) of the interdimer transition, and a small blue shift (1.8 meV) and an intensity decrease (2.9%) of the intradimer transition, we can approximately reproduce the $\Delta R(td = 0 \text{ ps})$ spectrum,". I suppose they analysed the data, interpreted the spectral changes in terms of shifts of band and intensities and then refined the spectral shifts and intensity changes. The data interpretation and analysis should be presented in a more scientific way, not "assuming" something but explaining why the data change should be interpreted in this way.

Reply to comment (9) of Reviewer #2

As for the analyses of the spectra of reflectivity changes ΔR , we reported the detailed methods and quantitative information in Supplementary Note 2. Here, let us briefly summarize the content of Supplementary Note 2 below.

First, we reproduced the steady-state reflectivity (R) and conductivity (σ) spectra with five Lorentz oscillators expressed by eq. (S2). The oscillators 1 and 2 are interdimer and intradimer transitions, respectively. The other three oscillators 3-5 are intramolecular vibrations. The R and σ spectra are well reproduced by the red lines as shown in Figs. 2a,b. After that, the ΔR spectra are reproduced by adjusting the energy positions and the oscillator strengths of the oscillators 1 and 2. The changes of the parameters are listed in Supplementary Table 3. Please see Supplementary Note 2. The physical meanings of the changes of parameters are discussed in the main text from line 1 in page 9 to line 3 in page 10.

Other minor points:

Comment (10) of Reviewer #2

- p15 the authors mention " SHG are not observed in κ -Cl down to 10 K. This indicates that the inversion symmetry continues to exist at a macroscopic scale below 40 K." Did the authors track SHG for light polarization // c only, or did they try perpendicular to c as well? Indeed polar order

could appear in different directions depending on the nature of the symmetry breaking.

Reply to comment (10) of Reviewer #2

We tried to detect SHG for the polarization of the incident light $\perp c$ as well as $//c$. However, no SHG signals were observed. If the inversion symmetry is lost in the steady state, some signals proportional to the electric field of a terahertz pulse should be observed as demonstrated in ferroelectrics in several previous papers (e.g. T. Miyamoto *et al.*, Nature Communications **4**, 2586 (2013), and H. Yamakawa *et al.*, Scientific Reports **6**, 20571 (2016)). In our experiments, however, such signals were never observed, supporting that there is still no inversion symmetry below 40 K.

We modified the related discussion as follows.

[Line 6-8 in page 6 of the new MS]

“In both κ -Cl and κ -CN, no SHG signals were detected in the steady states at all the temperatures from 294 K to 10 K irrespective of the polarization direction of the incident pulse in contrast to α -(ET)₂I₃.”

Comment (11) of Reviewer #2

- p6 it is written " $\Delta I_{\text{SHG}}(td)$ increases in accord to $[E_{\text{THz}}(td)]^2$ (the red line in the middle panel) without delay, indicating that the initial response is electronic in nature ". However, there are some intra-dimer/molecular modes, like molecular bending, which may carry a dipole. Such intra-molecular bending mode may fall in the 100 cm⁻¹ range, with ¼ period shorter than 75 fs. Therefore I don't understand why the contribution of molecular motion can be ruled out.

Reply to comment (11) of Reviewer #2

In our experiments, we use the probe pulses with the temporal width of 90 fs, which corresponds to the time resolution. The frequency of 100 cm⁻¹ corresponds to the period of 0.33 ps, the quarter of which is about 80 fs. On the other hand, the change of the square of the electric field from zero to its maximum occurs with about 200 fs. In our experimental condition, therefore, the rapid response within 100 fs is difficult to measure precisely. In this context, contributions of molecular motions as well as atomic motions cannot be ruled out. On the other hand, it is natural to consider that the initial rapid response along the electric field of the terahertz pulse is ascribed to the field-induced intradimer charge transfers and all the other changes such as atomic or molecular motions follows them. In addition, the coherent oscillation attributed to the dimerization mode with the period of 1.15 ps is clearly observed and the delayed response clearly shows the rise time of 0.28 ps, which is close to the quarter of the period (1.15 ps) of the dimerization mode. Therefore, we believe that our interpretation is reasonable.

Comment (12) of Reviewer #2

- The figure are small and hard to read. Especially the figures showing the molecular structures and

packing (1b, 2h, 5c). It would be much easier to read with molecules (or dimers) represented by strings (or pairs of strings) like in Gati's PRL for example (120, 247601, 2018). The molecular view with atoms too small to be distinguished is useless. Or a larger figure must be used (for the first one may be). It will be much easier to read the CO as well, without atomic balls superposing to CO balls.

Reply to comment (12) of Reviewer #2

We thank Reviewer #2 for his/her thoughtful comment on the figures. We think that for general readers we had better use the same style of figures of molecular arrangements throughout the paper. Namely, we would like to avoid mixing both schematic and actual structural figures.

Taking the comment into account, we modified Fig. 1, in which the schematic figure (Fig. 1b) was expanded so that the molecular structures and packing could be seen more easily.

Comment (13) of Reviewer #2

- It is mentioned in the text p 5 "we set the photon energy of the incident pulse to be 0.95 eV, since κ -Cl and κ -CN are almost transparent for this pulse ". However, in sup. Info. It is written that the absorption depth is about 1 μ m

Reply to comment (13) of Reviewer #2

We thank Reviewer #2 for his/her careful reading of our paper. The comment is quite reasonable. We corrected the related description as follows.

[Line 1-3 in page 6 of the new MS]

"In the SHG measurements, we set the photon energy of the incident pulse to be 0.95 eV, since the absorption depth at 0.95 eV is enough long as compared to the coherence length of the reflection-type SHG in κ -Cl and κ -CN."

[Line 19-21 in page 7 in the new MS]

"The details of the experimental conditions of the SHG measurements and the estimations of the polarizations are reported in Supplementary Note 1."

Comment (14) of Reviewer #2

- in equation (1), for the fit of the data, the oscillating part is written with a "c" subscript, while in the text it is an "osc" subscript. Please check consistency. In the text Tau_d is also mentioned and it should be more clearly defined.

Reply to comment (14) of Reviewer #2

We thank Reviewer #2 again for his/her careful reading of our paper. We corrected the subscripts of parameters in eq. (1) as follows.

[eq. (1) in line 11 in page 11 of the new MS]

$$\tau_C \rightarrow \tau_{\text{OSC}}$$

$$\omega_C \rightarrow \omega_{\text{OSC}}$$

$$\phi_C \rightarrow \phi_{\text{OSC}}$$

In addition, taking the comment of Reviewer #2 into account, we modified the description about the definition of τ_d as follows.

[Line 1-9 in page 14 of the new MS]

“We show the electric-field dependences of the maximum values of $|\Delta R(t_d)/R|$, $\max|\Delta R(t_d)/R|$, in Fig 3e, the values of A_α , A_β and A_γ which characterize the magnitudes of three processes $\alpha - \gamma$ in Fig. 3f, and the values of τ_2 which dominate the initial decay of the electric-field-induced polar state in Fig. 3g. More precisely, the decay dynamics is reproduced by the sum of two exponential terms with the shorter decay time τ_2 and the longer decay time τ_3 . In order to understand the electric-field dependence of the decay time of the polar state more clearly, we defined the time until the $|\Delta R(t_d)/R|$ signal becomes half the $\max|\Delta R(t_d)/R|$ as the effective decay time τ_d and plotted it in Fig. 3g.”

Comment (15) of Reviewer #2

- p12 " The parameter values adopted are..." error bars should be given for the values extracted from the fit. A table listing the results of the fits in sup info would help also.

Reply to comment (15) of Reviewer #2

Taking the comment of reviewer #2, we added error bars in the parameter values shown in page 12, in Table S2, and in Table S3.

Comment (16) of Reviewer #2

- p14 when authors mention " the potential barrier seems to be very small". What is the order of magnitude expected compared to thermal or electric energy?

Reply to comment (16) of Reviewer #2

The metastable charge-ordered state returns to the Mott insulator on the picosecond time scale even at 40 K. Therefore, it seems that the barrier from the charge-ordered state to the Mott insulator state is comparable to 40 K. Anyway, we think that the following description in the old MS was ambiguous and deleted it in the new MS.

“while the potential barrier seems to be very small”

Then, the related description was modified as follows.

[Line 19-20 in page 14 in the new MS]

“This indicates the presence of a finite potential barrier between the paraelectric Mott insulator phase and the polar CO phase, and the polar CO is a metastable state.”

Comment (17) of Reviewer #2

The writing and language should be improved. Some sentences are not fluent, hard to read due to wording or typos, like

"much attention not only from as a new topic in solid state physics"

"rapidly decreases to zero at when the electric field crosses zero"

" As the electric-field amplitude increases, both of the initial decrease of the reflectivity and its recovery become slow." -> slows down?

Parallel should be written "||" "not "||"

In sup info $\Delta P 2\kappa - C1$ should be written $(\Delta P \kappa - C1)^2$

Reply to comment (17) of Reviewer #2

We corrected the descriptions which Reviewer #2 commented on. In addition, we modified some sentences to improve the readability.

Replies to comments of Reviewer #3

We thank Reviewer #3 for his/her careful reading of our paper and valuable comments. We thank Reviewer #3 for recognizing the value of our paper and saying “The paper is scientifically sound, clearly presented, its topic should be of broad interest and definitely stimulate further theoretical and experimental work.”.

We divided the comments of Reviewer #3 to four parts (1)-(4). We show the reply to each comment below.

Comment (1) of Reviewer #3

I have some questions concerning the analysis of the temperature dependence: The analysis in Fig. 5 is based on the maximum reflectivity change $\Delta R/R$. The later seems to be mostly dominated by process “alpha” in Fig.4, i.e., the instantaneous response to the THz pulse. On the other hand, Fig 3f shows that process “alpha” is proportional to the square of the THz field down to low THz amplitudes, and therefore should relate to the equilibrium properties of the material rather than a non-equilibrium phase.

Reply to comment (1) of Reviewer #3

We thank Reviewer #3 for his/her valuable comment. The measurements of the time characteristics of the reflectivity changes, $\Delta R(t_d)/R$, at various temperatures shown in Fig. 4 were performed using the terahertz pulse with the maximum electric field of 288 kV/cm. As seen in Figs. 4a and d, the long-lived component in the reflectivity change increases with decrease of temperature from 290 K to 40 K. To see this fact clearly, we prepared the right figure, in which we compare the time characteristic of $\Delta R(t_d)/R$ at 290 K (the upper panel) shown in Figs. 4d and that at 40 K (the lower panel) shown in Fig. 3d.

The time characteristic of $\Delta R(t_d)/R$ at 290 K shown in Figs. 4d is very similar to that of the α component of $\Delta R(t_d)/R$ at 40 K shown by the blue line in Fig. 3d. Therefore, we can consider that $\Delta R(t_d)/R$ at 290 K is dominated by the α component, which is attributed to the electric-field-induced charge transfer within each dimer. The magnitude of this component at 40 K is about 14 times as large as that at 290 K. In addition, at 40 K, the β component becomes dominant, which is much

larger than the α component. The β component is attributed to the delayed response associated with the stabilization process of the polar CO state by molecular displacements and shows the nonlinear dependence on the electric field (Fig. 3f). Thus, the enhancement of $\max|\Delta R(t_d)/R|$ with decrease of temperature down to 40 K is mainly ascribed to the β component, although the increase of the α component also contributes to the enhancement of $\max|\Delta R(t_d)/R|$.

To make this point clearer, we modified the related discussions as follows.

[Line 8-20 in page 15 of the new MS]

“The temperature dependence of the maximum values of $|\Delta R(t_d)/R|$, $\max|\Delta R(t_d)/R|$, is shown in Fig. 5a. With the decrease in the temperature from 290 K, $\max|\Delta R(t_d)/R|$ increases; it attains the maximum value at 40 K. Simultaneously, the rise time of $|\Delta R(t_d)/R|$ increases and the decay time of $|\Delta R(t_d)/R|$ is prolonged as seen in Fig. 4d. This suggests that the β component reflecting the stabilization process of the polar CO state is enhanced with the decrease in the temperature. In fact, the time characteristic of $\Delta R(t_d)/R$ at 290 K shown in Fig. 4d is very similar to that of the α component of $\Delta R(t_d)/R$ at 40 K shown in Fig. 3d and the β component does not exist at 290 K. All these results indicate that at approximately 40 K, the instability to the polar CO state is maximized and the electric-field-induced CO is most likely to be stabilized. By using the value of τ_d obtained from the decay dynamics of $\Delta R(t_d)/R$, we can see that the life time of the polar CO state increases with decrease of temperature down to 40 K, e.g., $\tau_d \sim 0.44$ ps at 290 K, 0.78 ps at 100 K, and 1.7 ps at 40 K.”

Comment (2) of Reviewer #3

This would indicate that the 40K temperature scale has a meaning already for the equilibrium properties. Is this correct?

Reply to comment (2) of Reviewer #3

This comment is closely related to comment (1) of Reviewer #3. As mentioned in the reply to comment (1), the increase of $\max|\Delta R(t_d)/R|$ is mainly dominated by the increase of the β component. The results suggest that at approximately 40 K, the instability to the polar CO state is maximized, and the electric-field-induced CO is most likely to be stabilized.

Comment (3) of Reviewer #3

Later the authors discuss a picture in which the low T phase may be interpreted as ordered domains. Would that mean that the 40K scale is related to the onset of the domain order, different from the 27K phase transition?

Reply to comment (3) of Reviewer #3

We thank Reviewer #3 for asking us a confirmation question. The answer is yes. The reflectivity change, $|\Delta R(t_d)/R|$, decreases sharply below 27 K. At this temperature, it is natural to consider that the domain order is grown to some extent. At 40 K, the instability to the polar CO is maximized. In this sense, we can consider that 40 K corresponds to the onset of the domain order. The related discussion is given in line 1-4 in page 17 of the new MS as follows.

“Above 40 K, the system is a homogeneous Mott insulator and the applied electric field induces a transient polar CO as illustrated in Fig. 2h. With the decrease in temperature below 40 K, the charge disproportionation in each dimer increases in the steady state as shown in the upper part of Fig. 5c.”

Comment (4) of Reviewer #3

The beta process, on the other hand, has a clearly nonlinear dependence of the field, with different regimes at large and small fields. To characterize the temperature-dependence of the non-equilibrium transition, would it be possible to provide both an analysis of the temperature dependence of the alpha process and the “beta“ process?

Reply to comment (4) of Reviewer #3

We thank Reviewer #3 for his/her valuable question. This question is also related to comments (1) and (2). As stated in the reply to the comment (1), the reflectivity change at 290 K does not include the β component but includes only the α component. At 40 K, the magnitude of the α component is about 14 times as large as that at 290 K. In addition, the magnitude of the β component is more than twice as large as that of the α component as seen in Fig. 3d. Thus, both of the α and β components are enhanced, while the enhancement of the β component seems to be more significant. We did not analyze the time characteristics of the reflectivity changes above 40 K. It is because with increase of temperature, the signal magnitudes become small, which makes the precise analyses difficult. Judging from the temperature dependence of the time characteristics of $\Delta R(t_d)/R$ shown in Figs. 4a and d, with the decrease in temperature, the α component seems to increase monotonically, while the β component becomes prominent below 80 K.

REVIEWER COMMENTS

Reviewer #1 (Remarks to the Author):

As I said previously, I think the research done is of nice quality. I probably agree with a lot of the conclusions. I still, however, think the manuscript does not provide clear, well-justified explanations how the measurements prove the conclusions. This is a serious enough problem to stop me from recommending publication.

1. Figure 1b presents a cartoon depicting a very specific purely electronic mechanism for the polarization alongside a line labeled ETHz. This is essentially an assertion of the papers main conclusion made before any data is shown (maybe this was meant as a hypothesis -- this is unclear). As a referee, especially for a high impact journal, I need to see direct and clear arguments how the paper reaches each conclusion. The paper also needs to be understandable upon reading the text once in order. Reading forward, there are some arguments to support the conclusions, but they are not made particularly well made or clear. A lot of work seems to rest on a silent assumption that the transient state is the same as closely related ground states of the compounds in question. This is not unreasonable in itself, but the manuscript needs to make direct arguments.
2. In lines 175-179 the text describes how the spectral changes can be justified in terms of changes of different energy levels within the systems. It does not address why the purported charge transfer would cause these energy level changes. In particular I would expect that the proposed mechanism would primarily be reflected in a broadening? Without this, I feel the changes are more "parameterized" rather than "explained" as claimed in the text.
3. In line 185 the text says "A likely origin of such a delayed response is the decrease in the dimerization in each dimer by electric-field-induced molecular displacements, as shown in the bottom part of Fig. 2h, which decreases t and U ." I agree this assignment is plausible, but it would be better to either explain why it is likely or to use weaker language such as a "possible" explanation. Saying that decreased dimerization "decreases" t_1 is a tautology. This would be better removed.
4. The explanation around line 191 is also something I cannot be sure is correct. It contains apparent circular reasoning and possible self-inconsistencies. It seems to invoke the blue shift as both the premise and the conclusion. In addition, I thought that t_1 causes the dimerization split? If this is reduced, will this not cause a red shift to the intradimer transition? The logic of what is presented is not clear enough for me to evaluate this section.
5. In terms of the long-timescale effects reported, the manuscript demonstrates a maximum timescale of ~ 2 ps and a maximum investigated delay of 10 ps. I think this is insufficient to call the effect metastable without some clear argument or other evidence. This claim, on line 299, is already preceded by calling the state "long lived". "Long lived" is probably enough and more proportionate to the evidence provided.
6. In line 348 the text assigns the observed enhanced signal to increased charge disproportionation. This is reasonable, but couldn't one equally well assume that a larger fraction of dimers have moments or possibly that more dimer moments are aligned with one-another? The paper references possibilities for inhomogeneity later. Would it not be better to simply and exclusively report the conclusion in terms of the average order parameter? This is the only measured quantity.
7. A large fraction of the discussion seems very heavily (maybe exclusively) predicated on prior conclusions of equilibrium studies. I feel that the text still relies to an excessively large extent on an unstated assumption that the transient state is the same as equilibrium ferroelectric states in related materials. It also devotes a lot of space to the lack of signal in κ -(ET)₂Cu₂(CN)₃ even though it's very difficult to make a strong statement about this – we simply do not have tools to address the

detailed information invoked. My opinion/advice is that it would be better to be briefer and revisit the general issues involved in the introduction .

8. While I do agree that the SHG signal comes from polarization, the manuscript still does a mediocre job of conveying this to the reader. It essentially just asserts that SHG measures polarization, which is not precisely true here and SHG can, in principle arise in other ways. Some further details are in the supplementary, but this is not referred to.

Reviewer #2 (Remarks to the Author):

Updated report:

The authors have satisfactorily dealt with some of my criticisms and comments.

However, the way the manuscript is written is still ambiguous, with data interpreted as due to polar CO from the very beginning. Authors should start by presenting the results (SHG and reflectivity) then explain that a polar state is identified (SHG) and then interpret the results as the formation of polar state driven by THz electric field, which may then be interpreted as resulting from polar CO, based on the response of similar materials or based on theory. Additional rewriting is necessary, with additional data/discussion to support the claims or interpretation.

Additional comments:

1 For example, P6 : the *c*-axis. $\Delta I_{\text{SHG}}(t_d)$ increases in accord to $[E_{\text{THz}}(t_d)]^2$ (the red line in the middle panel) without delay, indicating that the initial response is electronic in nature and that a charge disproportionation is produced. "Without delay" is very vague. In addition, this should be "without delay within the experimental resolution". It should be explained why only CO plays a role and not molecular motions. Changes occurring within 150 fs may involve molecular dimerization mode as this may fall in the $\frac{1}{4}$ period of the mode. This is of importance since the authors underline the fact that "molecular displacements stabilize the CO in another ET-based molecular compound, α -(ET)₂I₃ (Ref. 16)." and given that the changes of reflectivity are maximized around 500 fs around which the electronic change is completed. It is then very likely that the shift in energy of the intra and inter-dimer bands, related to a splitting of the bonding and anti-bonding orbitals, is associated with atomic motions.

2 Another key question is to show how the SHG signal changes with the amplitude of the THz field. Fig. 1e must be shown for different amplitudes of the THz field $E_{\text{THz}}(0)$ to show that the response scales with to $[E_{\text{THz}}]^2$. Only the dependence of reflectivity with E_{THz} is shown in Fig. 3. It is important to show how the time dependent SHG signal changes with E_{THz} by showing at least the SHG signal at low and high THz fields (below 40 and above 200 kV/cm).

It would be important also to show the amplitude of the SHG signal after long time like 5 ps as function of E_{THz} to see if the signal changes like $[E_{\text{THz}}]^2$ or not and to evidence a threshold in the formation of the "long lived" polar state.

In addition, the answer to my comment (2) is not satisfactory. Reflectivity change is associated with a different electronic state and not directly related to polar order (but may be coupled to). The technique, which is measuring the polar order, is the SHG. Therefore, the polar CO can't be monitored by reflectivity directly except if some correlation exist.

The comment lines 195-199

"In κ -Cl at 40 K, the time evolution of $\Delta I_{\text{SHG}}(t_d)$ (the middle panel of Fig. 1e) reflecting the electric-field-induced polarization and that of $\Delta R(t_d)$ at 0.5 eV (the lower panels of Fig. 2e) associated with the intradimer transition reflecting the electric-field-induced intermolecular charge transfers in each dimer are almost in agreement with each other."

or in note 3

"time characteristics of $\Delta I_{\text{SHG}}(td)$ and $-\Delta R(td)/R$ are almost the same with each other "

are not strong arguments but just observations. If this can't be explained, it should be stated that authors assume to probe indirectly polar order through the correlation is reflectivity change to explain why all the analysis of polar CO can be made from reflectivity change. Is reflectivity change related to SHG by the physical process? What can explain such an assumption? Are SHG and reflectivity change proportional or is one squared with respect to the other?

In addition, it seems to be that the decay of SHG and polar order occurs within 5 ps (from Fig. S1a), while the decay of electronic state probed by reflectivity is not completed after 10 ps. Therefore it seems to be that reflectivity and SHG are probing two different signals and are sensitive to two different processes: electronic state vs polar order.

It is also necessary to show SHG data up to 10 ps to show that the polar order is "long-lived" and that it is not the electronic state.

3 When the authors indicate "Then $\Delta I_{\text{SHG}}(t_d)$ remains after the electric field diminishes, indicating that the charge disproportionation state is metastable in κ -Cl, reminiscent of CO. The important feature is that the SHG signal does not decrease so much at the time when the terahertz electric field crosses zero and changes its sign at ~ 0.2 ps. This suggests that the polar state formed by the first positive peak of the terahertz electric field around the time origin is stabilized within ~ 0.3 ps and its polarization does not reverse by the negative electric fields from 0.25 ps to 0.75 ps."

It seems that the conclusion of the result is a bit fast. The authors should comment first the results: "Then $\Delta I_{\text{SHG}}(t_d)$ remains after the electric field diminishes. The important feature is that the SHG signal does not decrease so much at the time when the terahertz electric field crosses zero and changes its sign at ~ 0.2 ps."

Then they can comment on "polar state formed by an electric field "

And finally they can provide tentative interpretation and discussion.

The reminiscent SHG suggests that the polar state formed by the first positive peak of the terahertz electric field around the time origin is stabilized within ~ 0.3 ps and its polarization does not reverse by the negative electric fields from 0.25 ps to 0.75 ps. Given that SHG due to CO is observed in K-Cl

and theoretically predicted in KCl we tentatively assign the observed SHG to the charge disproportionation state is metastable in κ -Cl, reminiscent of CO.

For the reflectivity change, the presentation of the results and discussion is more convincing with presentations of the results, followed by an interpretation "The result is explained as follows " line 175.

Typos :

A scale is missing in Fig 3d for the alpha component

140-141 "These results indicate that the field-induced polar state is formed by an electric field but is unstable" This is redundant "These results indicate that the polar state is formed by an electric field but is unstable" is enough

105 enough long -> long enough

Reviewer #3 (Remarks to the Author):

The authors have convincingly clarified the minor issue raised in my report. As I have stated previously, the paper is interesting and will definitely stimulate further work. I recommend the revised manuscript for publication in Nature Communications.

Reply to comments of Reviewer #1

We thank Reviewer #1 for his/her careful reading of our revised manuscript (MS) and valuable comments and suggestions. We divide his/her comments into 9 parts (0)-(8), the replies to which are listed below.

Comment (0) of Reviewer #1

As I said previously, I think the research done is of nice quality. I probably agree with a lot of the conclusions. I still, however, think the manuscript does not provide clear, well-justified explanations how the measurements prove the conclusions. This is a serious enough problem to stop me from recommending publication.

Reply to comment (0) of Reviewer #1

We thank Reviewer #1 for his/her thoughtful comment. Considering this comment, we thoroughly revised our MS so that the explanations of the experimental results and the logic to the conclusion became clearer. The details of the revisions are shown in the replies to comments (1)-(8) of Reviewer #1 and also in those to comments (1)-(4) of Reviewer #2.

Comment (1) of Reviewer #1

Figure 1b presents a cartoon depicting a very specific purely electronic mechanism for the polarization alongside a line labeled ETHz. This is essentially an assertion of the papers main conclusion made before any data is shown (maybe this was meant as a hypothesis -- this is unclear). As a referee, especially for a high impact journal, I need to see direct and clear arguments how the paper reaches each conclusion. The paper also needs to be understandable upon reading the text once in order. Reading forward, there are some arguments to support the conclusions, but they are not made particularly well made or clear. A lot of work seems to rest on a silent assumption that the transient state is the same as closely related ground states of the compounds in question. This is not unreasonable in itself, but the manuscript needs to make direct arguments.

Reply to comment (1) of Reviewer #1

We thank Reviewer #1 for his/her thoughtful comment.

First, we deleted the middle panel of Fig. 1b and divided the original Fig. 1 into two figures, Fig. 1 and Fig. 2, in the revised MS. By this change, Figure 1 shows only crystal

structures and fundamental steady-state properties in κ -Cl and κ -CN. In Fig. 2, we show the results of the electric-field-induced SHG.

Second, we deleted the sentence at line 61-64 in page 4 and line 81-82 in page 5 and modified the sentence at line 91-91 and at 96-97 in page 5 in the old MS to simplify the introductory part. We believe that the introduction is improved.

Third, we divided Fig. 2 in the old MS into two figures, Fig. 3 and Fig. 4 in the revised MS, which show respectively the results of the electric-field-induced reflectivity changes and the cartoon for the explanation of the results of both SHG and reflectivity changes.

The revisions of the descriptions in the sections of Results and Discussions are mentioned in the other replies to comments of two Reviewers.

Comment (2) of Reviewer #1

In lines 175-179 the text describes how the spectral changes can be justified in terms of changes of different energy levels within the systems. It does not address why the purported charge transfer would cause these energy level changes. In particular I would expect that the proposed mechanism would primarily be reflected in a broadening? Without this, I feel the changes are more “parameterized” rather than “explained” as claimed in the text.

Reply to comment (2) of Reviewer #1

We thank Reviewer #2 for his/her important comment. We analyzed the spectral changes of reflectivity, $\Delta R/R$, very carefully. We summarize our analyses below.

We first analyzed the steady-state polarized reflectivity (R) and optical conductivity (σ) spectra shown in Fig. 3a, b in the revised MS (Fig. 2a, b in the old MS). The structure peaked at 0.43 eV and the broader structure around 0.2 eV observed in the σ spectrum are attributed to the intradimer transition indicated by the orange arrow in Fig. 3c in the revised MS and the interdimer transition indicated by the green arrow in Fig. 3c in the revised MS, respectively. The three sharp peaks observed at 0.109 eV, 0.157 eV and 0.164 eV are assigned to the intramolecular vibration (a_g) modes. Those modes become IR-active via electron intramolecular-vibration (EMV) coupling. These interpretations had been established by the previous studies (e.g. ref. 35). Taking those five kinds of optical absorption bands into account, we assumed that the complex dielectric constant consists of five Lorentz oscillators and analyzed the R and σ spectra. The fitting curves shown by the red lines in Fig. 3a,b in the revised MS (Fig. 2a,b in the old MS) reproduced well the experimental R and σ spectra. The parameter values used are listed with error bars

(one standard deviation) in Table S2 in the Supplementary Note 2 in Supplementary Information.

Next, we analyzed the changes of the reflectivity spectrum ΔR induced by the terahertz electric field shown in Fig. 3f,g in the revised MS. In the analysis, we tried to reproduce ΔR spectrum by changing the parameters of two Lorentz oscillators expressing the intradimer and interdimer transitions. The changed parameters are only two for each transition: the oscillator strength and the energy position. The spectral widths were unchanged, so that no broadening effects were considered. At $t_d = 0$ ps, the photon energy $\hbar\omega_1$ of the interdimer transition increases from 207.3 meV to 213.4 meV, i.e., by 6.1 meV. The photon energy $\hbar\omega_2$ of the intradimer transition increases from 436.0 meV to 437.8 meV, i.e., by 1.8 meV, and its oscillator strength decreases by 2.9%. However, at $t_d = 0.5$ ps, $\hbar\omega_1$ of the interdimer transition decreases from the original value of 207.3 meV to 201.6 meV, i.e., by 5.7 meV. $\hbar\omega_2$ of the intradimer transition increases from the original value of 436.0 meV to 440.0 meV, i.e., by 4 meV, and its oscillator strength decreases by 5.4% from the original value. The obtained parameter values are listed with error bars (one standard deviation) in Table S3 in the Supplementary Note 2 in Supplementary Information.

Finally, we tried to interpret the changes of the two parameters, the transition energy and the oscillator strength, of the two bands. Taking the comment of Reviewer #1 into account, we have carefully checked our interpretations and revised a part of the explanations about them, which are detailed below.

(a) The blue shift and decrease of the oscillator strength of the intradimer transition at $t_d = 0$ ps

The photon energy of the intradimer transition is mainly determined by the splitting of the bonding- and antibonding-orbital in each dimer as shown in Fig. 3c in the revised MS, which is approximated to $2t_1$ (t_1 : intradimer transfer integral) and equal to U_{dimer} in the Mott insulator state (e.g. refs. 24 and 35). When there is a difference of the site energy potential, 2Δ , between two molecules in a dimer, the splitting is increased to $2\sqrt{t_1^2 + \Delta^2}$ and the charge disproportionation occurs (Fig. 3d in the revised MS). Considering the possibility of the charge disproportionation deduced from the SHG signal by the terahertz electric field, it is natural to consider that this blue shift of the intradimer transition at $t_d = 0$ ps is due to the generation of the difference in the site energy potential by the terahertz electric field. We consider that this difference would originate not only from the electric field itself, which changes directly the site energy potential, but also from the energy gain in the charge-disproportionated state due to the long-range

Coulomb interactions between molecules belonging to different dimers. The generation of the difference in the site energy potential causes the decrease of the hybridization of the molecular orbitals of two molecules in each dimer, which should reduce the oscillator strength of the intradimer transition, as observed in the experiment.

(b) The blue shift of the interdimer transition at $t_d = 0$ ps

The optical transition energy or Mott-gap energy in a half-filled Mott insulator is determined by the balance of the on-site Coulomb repulsion energy U and the width δ of the upper and lower Hubbard bands. It increases with increase of U and with decrease of δ . In a dimer Mott insulator studied here, U_{dimer} is approximated to $2t_1$ and δ is determined by the interdimer transfer integrals t and t' . When a difference of the site energy potential, 2Δ , is induced in each dimer, the interdimer transition energy should increase since U_{dimer} increases from $2t_1$ to $2\sqrt{t_1^2 + \Delta^2}$. Assuming that δ is unchanged, the interdimer transition energy would increase by the terahertz electric field as observed in the experiment.

The explanation of the reflectivity changes $\Delta R(t_d = 0.5 \text{ ps})$ (Fig. 3g in the revised MS) is not so straightforward. Taking into account the interpretation of $\Delta R(t_d = 0 \text{ ps})$ shown above, we can explain the results of $\Delta R(t_d = 0.5 \text{ ps})$.

(c) The blue shift and decrease of the oscillator strength of the intradimer transition at $t_d = 0.5$ ps

The delayed further blue shift of the intradimer transition at $t_d = 0.5$ ps suggests that some structural changes would be involved in the dynamics. A possible structural change is the decrease of the dimerization or equivalently the increase in the intermolecular distance between two molecules forming a dimer. It is because the terahertz electric field should not only induce intradimer charge transfer and the resultant charge disproportionation, but also pull apart two charged molecules in each dimer, which would decrease the dimerization. The time scale of the delayed change is evaluated to be about 0.3 ps from the analysis of the time evolutions of ΔR reported in Supplementary Note 3, which is almost in agreement with the quarter of the period of the breathing mode of dimer. The previous theoretical studies indicate that the reduction of the dimerization, that is, the decrease of the intradimer transfer integral t_1 would enhance the instability to the CO and tend to increase the charge disproportionation in each dimer. In this case, the energy difference, 2Δ , between two molecules in each dimer might be increased possibly via the enhancement of the long-range Coulomb interactions between two molecules belonging to different dimers. In the experiment, the intramolecular transition

energy approximated to $2\sqrt{t_1^2 + \Delta^2}$ increases at $t_d = 0.5$ ps. This result shows that the increase of 2Δ overcomes the decrease of t_1 . The increase of 2Δ and also the decrease of t_1 should suppress the hybridization of π -orbitals between two molecules in each dimer, reducing the oscillator strength of the intradimer transition, which can explain the experimental result.

(d) The red shift of the interdimer transition at $t_d = 0.5$ ps

As discussed above, the decrease of the dimerization increases the intradimer transition energy approximated to $2\sqrt{t_1^2 + \Delta^2}$. Since $2\sqrt{t_1^2 + \Delta^2}$ corresponds to U_{dimer} , this change should increase the intermolecular transition energy. However, the experimental result shows that the interdimer transition energy decreases at $t_d = 0.5$ ps. To explain this red shift of the interdimer transition, we should consider another factor. A possible factor is the change of the bandwidth; molecular displacements corresponding to the release of dimerization might increase a transfer integral between two molecules belonging to the neighboring dimers. If those transfer integrals increase, the bandwidth would increase, resulting in the decrease of the interdimer transition energy as observed in the experiment. To demonstrate this interpretation, further theoretical studies about the electron-phonon interactions associated with the intermolecular transfer integrals should be necessary.

Considering the comment (2) of Reviewer #1, we added the above discussions from line 3 of page 7 to line 15 of page 10 in the revised Supplementary Note 2. In addition, we corrected the related discussion in the main text, which is shown below.

[Line 5-7 of page 8 in the revised MS]

“The energies of two transitions increase with increase of U_{dimer} , while the interdimer transition is more sensitive to the interdimer transfer integral or equivalently the bandwidth δ .”

[Line 21 of page 8 to line 13 of page 10 in the revised MS]

“The magnitudes of $\Delta R(t_d = 0 \text{ ps})$ are plotted in Fig. 3f; it exhibits a characteristic minus-plus-minus structure. By assuming a blue shift (1.8 meV) and an intensity decrease (2.9%) of the intradimer transition, and a blue shift (6.1 meV) of the interdimer transition, we can approximately reproduce the $\Delta R(t_d = 0 \text{ ps})$ spectrum, as shown by the blue broken line in Fig. 3f. The directions of the shifts of the two bands are shown by the arrows in the same figure. Those spectral changes are explained as follows (see Supplementary Note 2 for details). The electric field induces the site energy difference in each dimer. This change gives rise to the charge disproportionation along the electric-

field direction (Fig. 3d and Fig. 4a→b) and also increases the splitting of the bonding and anti-bonding orbitals, which causes the blue shift and the intensity decrease of the intradimer transition.

Figure 3g shows the ΔR spectrum at $t_d = 0.5$ ps, $\Delta R(t_d = 0.5 \text{ ps})$, in which ΔR below 0.2 eV rather increases. By assuming a blue shift (4.0 meV) and a further intensity decrease (5.4%) of the intradimer transition, and a red shift (5.7 meV) of the interdimer transition, we can approximately reproduce the $\Delta R(t_d = 0.5 \text{ ps})$ spectrum as shown by the red broken line in Fig. 3g. The directions of the shifts of the two bands are also shown by the arrows in the same figure. A possible explanation of such a delayed response is the decrease in the dimerization in each dimer as illustrated in Fig. 4b→c. The electric field not only induces the charge transfer in each dimer but also pull apart charge-disproportionated two molecules. The theoretical studies indicate that the decrease in t_1 in each dimer favors the CO, so that those molecular motions would make the polar CO more stabilized and its lifetime longer. The decrease in t_1 in each dimer tends to decrease the splitting of bonding- and anti-bonding orbitals, while the stabilization of the CO means that the site-energy difference between two molecules in each dimer is enhanced, which tends to increase the splitting of bonding- and anti-bonding orbitals. The experimental result shows the blue shift of the intradimer transition, so that the latter effect overcomes the former effect. In this case, both the decrease in t_1 and the increase in the orbital splitting should suppress the oscillator strength of the intradimer transition. At $t_d = 0.5$ ps the interdimer-transition rather shows a red shift. The blue shift of the intramolecular transition suggests the increase in the splitting of the bonding and anti-bonding orbitals corresponding to U_{dimer} , which cannot explain the red shift of the interdimer transition. A possible origin for the red shift is the increase in the bandwidth by the increase in t and t' through the molecular motions corresponding to the release of the dimerization, which is also detailed in Supplementary Note 2.

In κ -Cl at 40 K, the time evolution of $\Delta I_{\text{SHG}}(t_d)$ (Fig. 2c) reflecting the electric-field-induced polarization and that of $\Delta R(t_d)/R$ at 0.5 eV (Fig. 3e) associated with the electric-field-induced charge disproportionation in each dimer are almost in agreement with each other (see Supplementary Note 3). This shows that the dynamics of the electric-field-induced polar CO at 40 K can be discussed from the time evolutions of $\Delta R(t_d)/R$ at 0.5 eV.”

Comment (3) of Reviewer #1

In line 185 the text says “A likely origin of such a delayed response is the decrease in

the dimerization in each dimer by electric-field-induced molecular displacements, as shown in the bottom part of Fig. 2h, which decreases t and U .” I agree this assignment is plausible, but it would be better to either explain why it is likely or to use weaker language such as a “possible” explanation. Saying that decreased dimerization “decreases” t_1 is a tautology. This would be better removed.

Reply to comment (3) of Reviewer #1

Considering this comment, we revised the related sentences. Please see the reply to comment (2) of Reviewer #1.

Comment (4) of Reviewer #1

The explanation around line 191 is also something I cannot be sure is correct. It contains apparent circular reasoning and possible self-inconsistencies. It seems to invoke the blue shift as both the premise and the conclusion. In addition, I thought that t_1 causes the dimerization split? If this is reduced, will this not cause a red shift to the intradimer transition? The logic of what is presented is not clear enough for me to evaluate this section.

Reply to comment (4) of Reviewer #1

We thank Reviewer #1 for his/her important comment. We agree with Reviewer #1. Our explanation on the change of the reflectivity spectrum in the previous MS might be insufficient. We thoroughly revised the related discussions. We show in detail the revision in the reply to comment (2) of Reviewer #1.

Comment (5) of Reviewer #1

In terms of the long-timescale effects reported, the manuscript demonstrates a maximum timescale of ~ 2 ps and a maximum investigated delay of 10 ps. I think this is insufficient to call the effect metastable without some clear argument or other evidence. This claim, on line 299, is already preceded by calling the state “long lived”. “Long lived” is probably enough and more proportionate to the evidence provided.

Reply to comment (5) of Reviewer #1

We thank Reviewer #1 for his/her valuable comment. As Reviewer #1 suggested, the word “metastable” might be misleading. On the other hand, from the results of Figs. 5a-

d and 5g in the revised MS, it is clear that the lifetime of the transient state generated by the terahertz electric field becomes longer with increase of $E_{\text{THz}}(0)$. Considering this point, we modified the related sentences as follows.

[Line 296-299 of page 15 in the old MS]

“Above this electric field, the polar CO state is stabilized and long lived. This indicates the presence of a finite potential barrier between the paraelectric Mott insulator phase and the polar CO phase, and the polar CO is a metastable state.”

[Line 20 of page 14-line 2 of page 15 in the revised MS]

“Above this electric field, the polar CO state is more stabilized and its lifetime becomes relatively long. This suggests that a small but a finite potential barrier would be produced between the paraelectric Mott insulator state and the polar CO state.”

Comment (6) of Reviewer #1

In line 348 the text assigns the observed enhanced signal to increased charge disproportionation. This is reasonable, but couldn't one equally well assume that a larger fraction of dimers have moments or possibly that more dimer moments are aligned with one-another? The paper references possibilities for inhomogeneity later. Would it not be better to simply and exclusively report the conclusion in terms of the average order parameter? This is the only measured quantity.

Reply to comment (6) of Reviewer #1

We thank Reviewer #1 for his/her important comment. To answer this, let us start the discuss about the nature of the electronic state far below 40 K, e.g., at 10 K. If the system has a macroscopic polarization, an SHG signal should be observed in the steady state. In addition, the time evolution of the reflectivity change by the terahertz field should include a component proportional to the waveform of the terahertz electric field. These features were clearly observed in the ferroelectric organic compounds of α -(ET)₂I₃ (ref. 16) and TTF-CA (ref. 39). However, in κ -Cl, no SHG signal is observed in the steady state and no signals proportional to the electric-field waveform are observed in the reflectivity changes at all the temperatures. These results indicate that a macroscopically polar state is not produced in κ -Cl.

According to the dielectric measurements (ref. 9), at high temperatures above 50 K, no dielectric response is observed, while the response to the terahertz electric field is enhanced with decrease of temperature from 200 K to 40 K. The response to the electric

field observed in the terahertz excitation case should also be included in the dielectric measurements, while no signals are observed at 50 K. This means that the dielectric measurements cannot detect the electric-field-induced change of the charge disproportionation. This is because in the dielectric measurements, the quasi-static electric field applied to the sample is very weak and the charge disproportionation induced is negligibly small. Considering these facts, the response observed in the dielectric measurements is attributable mainly to the alignment of a dipole moment in each dimer.

From 40 K to T_c , the response to the terahertz electric field is decreased, while the dielectric response is increased (see Fig. 7a in the revised MS). Therefore, the former response cannot be ascribed to the alignment of the dipole moment in each dimer. The decrease of the response to the terahertz electric field from 40 K to T_c is attributable to the increase of the dipole moment, that is, the increase of the charge disproportionation in each dimer. In the dielectric measurements, the dielectric constant might be dominated not only by the magnitude of the dipole moment but also by the ease of alignment of each dipole moment.

Below T_c , the dielectric constant rather decreases. We suppose that the microscopic polar CO domains are fully grown around T_c , below which the direction of the polarization of each domain tends to be frozen and difficult to be controlled even by a quasi-static electric field. In these dielectric processes, anions might play important roles.

These discussions are a little bit complicated and not so important for general readers to understand the main part of our paper, that is, the charge and molecular dynamics in the terahertz-electric-field-induced conversion from the Mott insulator state to the polar charge order state. Therefore, we did not add all of them in the revised MS. Considering the comment of the Reviewer #1, we modified the related sentences as follows.

[Line 347-351 of page 17 in the old MS]

“With the decrease in temperature below 40 K, the charge disproportionation in each dimer increases in the steady state as shown in the upper part of Fig. 5c. This is observed as a divergent behavior of the dielectric constant around $T_c = 27$ K; meanwhile, no second-order nonlinear optical signals are observed even at 10 K.”

[Line 5-10 of page 17 in the revised MS]

“With the decrease in temperature below 40 K, the charge disproportionation in each dimer is expected to increase in the steady state as shown in the upper part of Fig. 7c. A divergent behavior of the dielectric constant around $T_c = 27$ K is attributable to the increase of the dipole moment in each dimer and the increased ease of its alignment by a quasi-static electric field. Meanwhile, no second-order nonlinear optical signals are

observed even at 10 K.”

Comment (7)-1 of Reviewer #1

A large fraction of the discussion seems very heavily (maybe exclusively) predicated on prior conclusions of equilibrium studies. I feel that the text still relies to an excessively large extent on an unstated assumption that the transient state is the same as equilibrium ferroelectric states in related materials.

Reply to comment (7)-1 of Reviewer #1

In κ -Cl, microscopically polar CO domains randomly oriented are grown around T_c and a macroscopically polar CO state is not formed even at 10 K. In contrast, above 40 K, we succeeded in producing macroscopically polar CO state in κ -Cl by a strong terahertz electric field. In this sense, a strong terahertz electric field creates a new state, which does not appear in the steady state. To state clearly this point, we modified the related sentences in the summary paragraph as follows.

[Line 14-17 of page 23 in the revised MS]

“In summary, in the present study, we demonstrated in a two-dimensional Mott insulator of an organic molecular compound, κ -(ET)₂Cu[N(CN)₂]Cl, that a polar charge order was created via collective intermolecular charge transfers by a strong terahertz electric-field pulse, which is a hidden state never stabilized in the steady state even at low temperatures.”

Comment (7)-2 of Reviewer #1

It also devotes a lot of space to the lack of signal in κ -(ET)₂Cu₂(CN)₃ even though it's very difficult to make a strong statement about this – we simply do not have tools to address the detailed information invoked. My opinion/advice is that it would be better to be briefer and revisit the general issues involved in the introduction.

Reply to comment (7)-2 of Reviewer #1

We thank Reviewer #1 for his/her important suggestion. Before the concrete discussion on κ -CN, we would like to discuss the general aspect of the electric-field-induced SHG. The electric-field-induced SHG is a kind of third-order optical nonlinearity, which is expressed by the following formula (for example, Butcher, P. N. & Cotter, D. The Elements of Nonlinear Optics (Cambridge University Press, Cambridge, 1990)).

$$P(2\omega) = \varepsilon_0 \chi^{(3)}(-2\omega; \omega, \omega, 0) E(\omega) E(\omega) E(0) \quad (R1)$$

$$I_{\text{SHG}}(2\omega) \propto [\chi^{(3)}(-2\omega; \omega, \omega, 0)]^2 [I(\omega)]^2 [E(0)]^2 \quad (\text{R2})$$

Here, $E(\omega)$ and $I(\omega)$ are an electric field and intensity of an incident probe pulse, respectively, $E(0)$ is a quasi-static electric field, and $P(2\omega)$ is a nonlinear polarization. $\chi^{(3)}(-2\omega; \omega, \omega, 0)$ is the third-order nonlinear susceptibility for the electric-field induced SHG. The intensity of the electric-field-induced SHG, $I_{\text{SHG}}(2\omega)$, is proportional to the square of $P(2\omega)$. As a result, $I_{\text{SHG}}(2\omega)$ is proportional to $[E(0)]^2$. The frequency of the terahertz pulse is much smaller than that of the incident light pulse for SHG and of the optical gap, so that we can consider it equal to zero. When a response is dominated by a purely electronic process, the framework expressed by eqs. (R1) and (R2) can be used to explain a terahertz-electric-field-induced SHG. In this case, a transient $I_{\text{SHG}}(t_d)$ is proportional to the square of $E_{\text{THz}}(t_d)$. We consider that the SHG induced by a terahertz electric field in κ -CN can be discussed with eqs. (R1) and (R2) since it is fundamentally dominated by an electronic part, and the contributions of molecular dynamics are relatively small. Therefore, it is possible to regard the SHG signal, $I_{\text{SHG}}(t_d)$, proportional to $[E_{\text{THz}}(t_d)]^2$ experimentally observed in κ -CN as a general response, which can be treated in the framework of the third-order optical nonlinearity, while electronic instability to a possible polar state might also be included to some extent in this third-order nonlinear optical response. On the other hand, when a phase transition or a structural change is induced by a terahertz electric field, a response should not follow the general relation of the third-order optical nonlinearity, $I_{\text{SHG}}(t_d) \propto [E_{\text{THz}}(t_d)]^2$. We think that this is the case of κ -Cl. Namely, only the initial electronic response would be proportional to the square of the terahertz electric field. Considering these aspects, we consider that the presentation of the results of SHG and reflectivity changes in κ -CN is important to make clear the specificity of the response to the terahertz electric field observed in κ -Cl.

To make clear the main point of the above discussion, that is, the important difference in the SHG signals of κ -CN and κ -Cl, we revised the descriptions about the SHG signal in κ -CN in the revised MS as follows.

[Line 4-11 of page 7 in the revised MS]

“The SHG signal shows a sharp peak around the time origin and then rapidly decreases to zero at the time when the electric field crosses zero. After that the SHG signal increases again under the presence of the negative electric field from 0.25 ps to 0.75 ps. In this case, it is natural to consider that the polarization reverses depending on the electric field direction. The SHG signal vanishes immediately after $E_{\text{THz}}(t_d)$ diminishes for $t_d >$

0.75 ps. The signal is roughly proportional to $[E_{\text{THz}}(t_d)]^2$ in all the time region, so that it can be regarded as a kind of third-order optical nonlinearity³⁴. Thus, in κ -CN, the polarization induced by an electric field is not stabilized. This is in contrast to the case of κ -Cl.”

In connection with this change, we added the following reference about the nonlinear optical response in solids in the revised MS.

34. Butcher, P. N. & Cotter, D. *The Elements of Nonlinear Optics* (Cambridge University Press, Cambridge, 1990).

We renumbered the following references.

In addition, we added the more detailed explanations about the steady-state SHG and the terahertz-electric-field-induced SHG in the revised Supplementary Information as follows, which we believe can help general readers understand the value of the SHG measurements.

[Line 13 of page 3-line 22 of page 4 in the revised Supplementary Note 1]

“The SHG in a non-centrosymmetric material is expressed with the second-order nonlinear susceptibility $\chi^{(2)}(-2\omega; \omega, \omega)$ as follows⁵.

$$P(2\omega) = \varepsilon_0 \chi^{(2)}(-2\omega; \omega, \omega) E(\omega) E(\omega) \quad (\text{S1})$$

$$I_{\text{SHG}}(2\omega) \propto [\chi^{(2)}(-2\omega; \omega, \omega)]^2 [I(\omega)]^2 \quad (\text{S2})$$

Here, $E(\omega)$ and $I(\omega)$ are an electric field and an intensity of an incident laser pulse, respectively, and ε_0 is the permittivity of vacuum. $P(2\omega)$ is a nonlinear polarization. In the ferroelectric material of α -I₃, $\chi^{(2)}(-2\omega; \omega, \omega)$ is proportional to the ferroelectric polarization $P_{\alpha\text{-I}_3}$. The SH-intensity can be expressed by $I_{\text{SHG}}(\alpha\text{-I}_3) = C_1 (P_{\alpha\text{-I}_3})^2$. C_1 is a proportional constant. The electric-field-induced SHG in a centrosymmetric material is expressed with the third-order nonlinear susceptibility $\chi^{(3)}(-2\omega; \omega, \omega, 0)$ as follows⁵.

$$P(2\omega) = \varepsilon_0 \chi^{(3)}(-2\omega; \omega, \omega, 0) E(\omega) E(\omega) E(0) \quad (\text{S3})$$

$$\Delta I_{\text{SHG}}(2\omega) \propto [\chi^{(3)}(-2\omega; \omega, \omega, 0)]^2 [I(\omega)]^2 [E(0)]^2 \quad (\text{S4})$$

Here, $E(0)$ is a quasi-static electric field. The frequency of the terahertz pulse is much smaller than that of the incident light pulse for SHG and of the optical gap, so that we can consider it equal to zero and $E(0) = E_{\text{THz}}(0)$. $\chi^{(3)}(-2\omega; \omega, \omega, 0) E(0)$ is proportional to the terahertz-electric-field-induced polarization ΔP , and the intensity of the electric-field-induced SHG, $\Delta I_{\text{SHG}}(2\omega)$, is proportional to the square of ΔP . Therefore,

$\Delta I_{\text{SHG}}(2\omega)$ can be expressed as $\Delta I_{\text{SHG}}(2\omega) = C_2(\Delta P)^2$. C_2 is also a proportional constant.

On the basis of the framework mentioned above, The ratio of the SH intensity $\frac{\Delta I_{\text{SHG}}(\kappa\text{-Cl})}{I_{\text{SHG}}(\alpha\text{-I}_3)} \left(\frac{\Delta I_{\text{SHG}}(\kappa\text{-CN})}{I_{\text{SHG}}(\alpha\text{-I}_3)} \right)$ is equal to the square of the ratio of the polarization $\frac{C_2(\Delta P_{\kappa\text{-Cl}})^2}{C_1(P_{\alpha\text{-I}_3})^2} \left(\frac{C_2(\Delta P_{\kappa\text{-CN}})^2}{C_1(P_{\alpha\text{-I}_3})^2} \right)$. Here, $\Delta I_{\text{SHG}}(\kappa\text{-Cl})$ ($\Delta I_{\text{SHG}}(\kappa\text{-CN})$) and $\Delta P_{\kappa\text{-Cl}}$ ($\Delta P_{\kappa\text{-CN}}$) are the intensity of the SHG and the magnitude of the polarization induced by the terahertz electric field in $\kappa\text{-Cl}$ ($\kappa\text{-CN}$), respectively. The maximum intensity of the SHG signal shown in Fig. 2c (2d) induced by a terahertz electric-field pulse of the amplitude 407 kV/cm in $\kappa\text{-Cl}$ at 40 K (in $\kappa\text{-CN}$ at 50 K) is approximately 1% (0.6%) of the intensity of the steady-state SHG signal in $\alpha\text{-I}_3$. Here, we assume that the values of the proportional constants C_1 and C_2 in $\alpha\text{-I}_3$ and $\kappa\text{-Cl}$ ($\kappa\text{-CN}$), respectively, are equal. It is reasonable because the magnitude of the CO gap in $\alpha\text{-I}_3$ is comparable to those of the Mott gaps in $\kappa\text{-Cl}$ and $\kappa\text{-CN}$. From $\frac{\Delta I_{\text{SHG}}(\kappa\text{-Cl})}{I_{\text{SHG}}(\alpha\text{-I}_3)} \sim 0.01$ and $\frac{\Delta I_{\text{SHG}}(\kappa\text{-CN})}{I_{\text{SHG}}(\alpha\text{-I}_3)} \sim 0.006$, we obtain $\frac{\Delta P_{\kappa\text{-Cl}}}{P_{\alpha\text{-I}_3}} \sim 0.1$ and $\frac{\Delta P_{\kappa\text{-CN}}}{P_{\alpha\text{-I}_3}} \sim 0.08$. Namely, 10% (8%) of the polarization in $\alpha\text{-I}_3$ is generated by a terahertz electric field in $\kappa\text{-Cl}$ ($\kappa\text{-CN}$).”

We also added the following reference about the nonlinear optical response in solids in the revised Supplementary Information.

5. Butcher, P. N. & Cotter, D. The Elements of Nonlinear Optics (Cambridge University Press, Cambridge, 1990).

We renumbered the following references.

Comment (8) of Reviewer #1

While I do agree that the SHG signal comes from polarization, the manuscript still does a mediocre job of conveying this to the reader. It essentially just asserts that SHG measures polarization, which is not precisely true here and SHG can, in principle arise in other ways. Some further details are in the supplementary, but this is not refereed to.

Reply to comment (8) of Reviewer #1

As Reviewer #1 commented on, there are several origins of a steady-state SHG. For example, even in a material with no ferroelectric polarization, SHG can be observed if it

has no inversion symmetry. In contrast, in an electric-field-induced SHG, origins would be more limited. An external electric field necessarily produces a finite polarization, even if it is very small. It is difficult to imagine a case in which an electric field does not produce any polarization but only break the inversion symmetry.

In the case of κ -CN, the time characteristic of the SHG signal is almost proportional to $[E_{\text{THz}}(t_d)]^2$, so it is natural to consider that this is mainly dominated by an electronic response and can be regarded as third-order optical nonlinearity as mentioned in the reply to comment (7)-2 of Reviewer #1. In other words, the nonlinear polarization is the origin of SHG. In κ -Cl, the SHG signal also rises up very fast. The rise time is much shorter than the time resolution as shown in the reply to comment (1) of Reviewer #2. We would like to ask Reviewer #1 to read that reply. From these facts, we consider that the initial rise of the SHG in κ -Cl is also attributable to the electronic response, that is, the polarization generation by the intradimer charge transfers. This electronic part can also be regarded as a kind of third-order optical nonlinearity mentioned above. In κ -Cl, however, the instability to the polar CO is large and the initial electronic response, that is, the charge disproportionation triggers the molecular motions stabilizing the polar CO. In our study, in the analyses of the time characteristics of the electric-field-induced reflectivity changes, we discriminate this structural change and the resultant enhancement of the polarization magnitude from the purely electronic part of the initial charge disproportionation or equivalently the third-order nonlinear response.

These discussions are somewhat complicated for general readers. On the other hand, it should be important to make clear the relation between the observed response to the electric field and the third-order nonlinearity. For this purpose, we added the detailed explanation about the electric-field-induced SHG from the viewpoint of the third-order optical nonlinearity in Supplementary Note 1 in the revised Supplementary Information as mentioned in the reply to comment (7)-2 of Reviewer #1. This is stated in the main text as follows.

[Line 14-17 of page 7 in the revised MS]

“The experimental conditions of the SHG measurements, the framework of the electric-field-induced SHG as the third-order optical nonlinearity, and the estimations of the electric-field-induced polarizations in κ -Cl and κ -CN are reported in Supplementary Note 1.”

In addition to the changes mentioned above, we shortened several sentences to simplify the explanations and discussions.

Reply to comments of Reviewer #2

We thank Reviewer #2 for his/her careful reading of our revised manuscript (MS) and valuable comments and suggestions. We divide his/her comments into 5 parts (0)-(5), the replies to which are listed below.

Comment (0) of Reviewer #2

The authors have satisfactorily dealt with some of my criticisms and comments. However, the way the manuscript is written is still ambiguous, with data interpreted as due to polar CO from the very beginning. Authors should start by presenting the results (SHG and reflectivity) then explain that a polar state is identified (SHG) and then interpret the results as the formation of polar state driven by THz electric field, which may then be interpreted as resulting from polar CO, based on the response of similar materials or based on theory. Additional rewriting is necessary, with additional data/discussion to support the claims or interpretation.

Reply to comment (0) of Reviewer #2

We thank Reviewer #2 for his/her thoughtful comment. Considering this comment, we thoroughly revised our MS so that the explanations of the experimental results and the logic to the conclusion became clearer. In particular, we added the discussions in the revised Supplementary Information to support the interpretation of the results shown in the main MS. The details of the revisions are shown in the replies to comments (1)-(4) of Reviewer #2 and also those to comments (1)-(8) of Reviewer #1.

Comment (1) of Reviewer #2

For example, P6 : the c-axis. Δ ISHG(td) increases in accord to $[E_{THz}(td)]^2$ (the red line in the middle panel) without delay, indicating that the initial response is electronic in nature and that a charge disproportionation is produced. "Without delay" is very vague. In addition, this should be "without delay within the experimental resolution". It should be explained why only CO plays a role and not molecular motions. Changes occurring within 150 fs may involve molecular dimerization mode as this may fall in the $\frac{1}{4}$ period of the mode. This is of importance since the authors underline the fact that "molecular displacements stabilize the CO in another ET-based molecular compound, α -(ET)₂I₃ (Ref. 16)." and given that the changes of reflectivity are maximized around 500 fs around which the electronic change is completed. It is then very likely that the shift in energy of the

intra and inter-dimer bands, related to a splitting of the bonding and anti-bonding orbitals, is associated with atomic motions.

Reply to comment (1) of Reviewer #2

The time characteristic of the SHG signal, $\Delta I_{\text{SHG}}(t_d)$, shown in Fig. 2c in the revised MS seem to include three components. The first component is the ultrafast rise of the signal around the time origin. To evaluate the rise time of the $\Delta I_{\text{SHG}}(t_d)$, we assume that it rises up with a time constant of τ_{rise} in proportion to $\left[1 - \exp\left(-\frac{t}{\tau_{\text{rise}}}\right)\right]$. For various τ_{rise} values, we calculated the time characteristics of this function convoluted with the pulsed component of $[E_{\text{THz}}(t_d)]^2$ around the time origin, which is shaded in yellow in Fig. R1 shown below. The calculated time characteristics are shown in the bottom panel of Fig. R1. The results show that τ_{rise} is much smaller than 0.1 ps, suggesting that the initial rise of the signal cannot be ascribed to the molecular motions with the time scale of 0.3 ps but can be attributed to purely electronic processes. The intradimer transfer energy is about 0.2 eV, which corresponds to the time scale of 20 fs. Therefore, it is reasonable to consider that the rise-up of the $\Delta I_{\text{SHG}}(t_d)$ signal is caused by the intradimer charge transfers.

The time resolution of our measurement system is mainly determined by the temporal width of the probe pulse, which is about 90 fs. By using the convolution analyses mentioned above, we can estimate the rise time of the signal even if its time constant is much shorter than 90 fs. We think that Fig. R1 and its explanation are somewhat complicated for general readers and not so important for them to understand the main content of our paper, so we did not show their details in the revised MS and Supplementary Information.

After such an ultrafast rise, $\Delta I_{\text{SHG}}(t_d)$ shows an additional increase up to 0.5 ps. This process is attributable to the molecular motions. As mentioned in the reply to comment (3) of Reviewer #2 below, it takes a very long time to measure the time evolution of the SHG signal, so that it is difficult to perform a systematic study of the field-induced SHG, such as an electric-field dependence and temperature dependence of the signal. In our study, instead, we use the reflectivity change at 0.5 eV. In fact, the time characteristics of ΔR at 0.5 eV and $\Delta I_{\text{SHG}}(t_d)$ are in good agreement with each other as shown in Fig. S1a in the revised Supplementary Information. The reason why the reflectivity change at this energy can give the same information as the $\Delta I_{\text{SHG}}(t_d)$ signal is explained in detail in the replies to comment (2) of Reviewer #1 and to comment (2)-1 of Reviewer #2.

Comment (2)-1 of Reviewer #2

Another key question is to show how the SHG signal changes with the amplitude of the THz field. Fig. 1e must be shown for different amplitudes of the THz field ETHz(0) to show that the response scales with to [ETHz]². Only the dependence of reflectivity with ETHz is shown in Fig. 3. It is important to show how the time dependent SHG signal changes with ETHz, by showing at least the SHG signal at low and high THz fields (below 40 and above 200 kV/cm).

It would be important also to show the amplitude of the SHG signal after long time like 5 ps as function of ETHz to see if the signal changes like [ETHz]² or not and to evidence a threshold in the formation of the "long lived" polar state.

In addition, the answer to my comment (2) is not satisfactory. Reflectivity change is associated with a different electronic state and not directly related to polar order (but may be coupled to). The technique, which is measuring the polar order, is the SHG. Therefore, the polar CO can't be monitored by reflectivity directly except if some correlation exist.

The comment lines 195-199

"In κ -Cl at 40 K, the time evolution of $\Delta I_{\text{SHG}}(t_d)$ (the middle panel of Fig. 1e) reflecting the electric-field-induced polarization and that of $\Delta R(t_d)$ at 0.5 eV (the lower panels of

Fig. 2e) associated with the intradimer transition reflecting the electric-field-induced intermolecular charge transfers in each dimer are almost in agreement with each other." or in note 3

"time characteristics of $\Delta I_{\text{SHG}}(t_d)$ and $\Delta R(t_d)$ are almost the same with each other " are not strong arguments but just observations. If this can't be explained, it should be stated that authors assume to probe indirectly polar order through the correlation is reflectivity change to explain why all the analysis of polar CO can be made from reflectivity change. Is reflectivity change related to SHG by the physical process? What can explain such an assumption? Are SHG and reflectivity change proportional or is one squared with respect to the other?

Reply to comment (2)-1 of Reviewer #2

We thank Reviewer #2 for his/her careful reading of our paper. In the SHG measurements, we cannot use the transmission configuration since both of the probe light and the second harmonic light are absorbed in the crystals used. Therefore, we adopted the reflection configuration, in which the SHG signals become very small as compared to those in the transmission configuration. It is because the coherence length of SHG in the reflection configuration is much shorter than that in the transmission configuration. The details of the experimental conditions of SHG are reported in Supplementary Note 1. In addition, we should avoid the carrier generations by the probe pulse, so that we cannot increase the photon density of the probe pulse to enhance the signal to noise ratio in the SHG measurements. To obtain the data of the electric-field-induced SHG in κ -Cl shown in Fig. 2c in the revised MS, we had to accumulate the signal for about one week (more than 150 hours). Therefore, it is difficult to perform a systematic study such as electric-field dependence of the dynamics of SHG although the reproducibility of the SHG data had been carefully ascertained. In this situation, it is impossible to obtain the data for E_{THz} below 40 kV/cm. We would like to emphasize that the magnitude of the electric-field-induced polarization is not small, which reaches about 10% of the steady-state polarization $1 \mu\text{C}/\text{cm}^2$ in α -I₃ at 10 K as detailed in Supplementary Note 1 and the drastic phenomenon is really induced by the terahertz electric field in κ -Cl. It is only a problem that the experimental condition is very severe.

In κ -CN, the situation is essentially the same as that in κ -Cl and it is difficult to obtain the SHG signal with good signal to noise ratio. However, in κ -CN, the electric-field-induced SHG signal shows a different behavior from that observed in κ -Cl. The time characteristic of $\Delta I_{\text{SHG}}(t_d)$ almost follows the square of $E_{\text{THz}}(t_d)$ as shown in Fig. 2d in the revised MS. This suggests that the SHG signal is proportional to the square of the

terahertz electric field at all the time region.

In κ -Cl, the overall time characteristic of SHG does not follow the square of $E_{\text{THz}}(t_d)$, so that it is difficult to discuss the electric-field-dependence of the magnitude of SHG signals. To discuss the electric-field dependence of the dynamics of polar CO state, we use the reflectivity change at 0.5 eV. To demonstrate that the time characteristic of the reflectivity change at 0.5 eV can give the same information as that of the SHG signal, we should clarify the origin of the reflectivity change at that energy. The detailed explanations about the origin of the reflectivity changes are given in the reply to comment (2) of Reviewer #1 and also at line 3 of page 7 to line 15 page 10 in the revised Supplementary Note 2. We would like to ask Reviewer #2 to read them.

Considering the comment (2)-1 of Reviewer #2, we corrected the related discussion in the main MS as follows.

[Line 21 of page 8 to line 13 of page 10 in the revised MS]

“The magnitudes of $\Delta R(t_d = 0 \text{ ps})$ are plotted in Fig. 3f; it exhibits a characteristic minus-plus-minus structure. By assuming a blue shift (1.8 meV) and an intensity decrease (2.9%) of the intradimer transition, and a blue shift (6.1 meV) of the interdimer transition, we can approximately reproduce the $\Delta R(t_d = 0 \text{ ps})$ spectrum, as shown by the blue broken line in Fig. 3f. The directions of the shifts of the two bands are shown by the arrows in the same figure. Those spectral changes are explained as follows (see Supplementary Note 2 for details). The electric field induces the site energy difference in each dimer. This change gives rise to the charge disproportionation along the electric-field direction (Fig. 3d and Fig. 4a→b) and also increases the splitting of the bonding and anti-bonding orbitals, which causes the blue shift and the intensity decrease of the intradimer transition.

Figure 3g shows the ΔR spectrum at $t_d = 0.5 \text{ ps}$, $\Delta R(t_d = 0.5 \text{ ps})$, in which ΔR below 0.2 eV rather increases. By assuming a blue shift (4.0 meV) and a further intensity decrease (5.4%) of the intradimer transition, and a red shift (5.7 meV) of the interdimer transition, we can approximately reproduce the $\Delta R(t_d = 0.5 \text{ ps})$ spectrum as shown by the red broken line in Fig. 3g. The directions of the shifts of the two bands are also shown by the arrows in the same figure. A possible explanation of such a delayed response is the decrease in the dimerization in each dimer as illustrated in Fig. 4b→c. The electric field not only induces the charge transfer in each dimer but also pull apart charge-disproportionated two molecules. The theoretical studies indicate that the decrease in t_1 in each dimer favors the CO, so that those molecular motions would make the polar CO more stabilized and its lifetime longer. The decrease in t_1 in each dimer tends to decrease the splitting of bonding- and anti-bonding orbitals, while the stabilization of the CO

means that the site-energy difference between two molecules in each dimer is enhanced, which tends to increase the splitting of bonding- and anti-bonding orbitals. The experimental result shows the blue shift of the intradimer transition, so that the latter effect overcomes the former effect. In this case, both the decrease in t_1 and the increase in the orbital splitting should suppress the oscillator strength of the intradimer transition. At $t_d = 0.5$ ps the interdimer-transition rather shows a red shift. The blue shift of the intramolecular transition suggests the increase in the splitting of the bonding and anti-bonding orbitals corresponding to U_{dimer} , which cannot explain the red shift of the interdimer transition. A possible origin for the red shift is the increase in the bandwidth by the increase in t and t' through the molecular motions corresponding to the release of the dimerization, which is also detailed in Supplementary Note 2.

In κ -Cl at 40 K, the time evolution of $\Delta I_{\text{SHG}}(t_d)$ (Fig. 2c) reflecting the electric-field-induced polarization and that of $\Delta R(t_d)/R$ at 0.5 eV (Fig. 3e) associated with the electric-field-induced charge disproportionation in each dimer are almost in agreement with each other (see Supplementary Note 3). This shows that the dynamics of the electric-field-induced polar CO at 40 K can be discussed from the time evolutions of $\Delta R(t_d)/R$ at 0.5 eV.”

With these changes, we hope Reviewer #2 would accept that the dynamics of the electric-field-induced polar CO can be discussed using the time characteristic of $\Delta R(t_d)$ at 0.5 eV.

Comment (2)-2 of Reviewer #2

In addition, it seems to be that the decay of SHG and polar order occurs within 5 ps (from Fig. S1a), while the decay of electronic state probed by reflectivity is not completed after 10 ps. Therefore it seems to be that reflectivity and SHG are probing two different signals and are sensitive to two different processes: electronic state vs polar order.

It is also necessary to show SHG data up to 10 ps to show that the polar order is "long-lived" and that it is not the electronic state.

Reply to comment (2)-2 of Reviewer #2

As mentioned in the reply to comment (2)-1 of Reviewer #2, it is difficult to obtain the SHG signal with good signal-to-noise ratio. To improve the signal to noise ratio, we limited the temporal range of the measurements. Although the measurement range is limited, we think that Fig. S1 in the Supplementary Information shows that the time

characteristics of $\Delta I_{\text{SHG}}(t_d)$ and $\Delta R(t_d)$ at 0.5 eV are almost the same with each other.

In the previous MS, we used the word “long-lived”, with which we would like to say that the lifetime of the electric-field-induced polar state is relatively long as compared with that observed in κ -CN. However, this word might be misleading. In the same sense, the word “metastable” used in the previous MS might not be appropriate. Therefore, in the new MS, we deleted these words.

Comment (3) of Reviewer #2

When the authors indicate "Then $\Delta I_{\text{SHG}}(t_d)$ remains after the electric field diminishes, indicating that the charge disproportionation state is metastable in κ -Cl, reminiscent of CO. The important feature is that the SHG signal does not decrease so much at the time when the terahertz electric field crosses zero and changes its sign at ~ 0.2 ps. This suggests that the polar state formed by the first positive peak of the terahertz electric field around the time origin is stabilized within ~ 0.3 ps and its polarization does not reverse by the negative electric fields from 0.25 ps to 0.75 ps."

It seems that the conclusion of the result is a bit fast. The authors should comment first the results: "Then $\Delta I_{\text{SHG}}(t_d)$ remains after the electric field diminishes. The important feature is that the SHG signal does not decrease so much at the time when the terahertz electric field crosses zero and changes its sign at ~ 0.2 ps."

Then they can comment on "polar state formed by an electric field "

And finally they can provide tentative interpretation and discussion.

The reminiscent SHG suggests that the polar state formed by the first positive peak of the terahertz electric field around the time origin is stabilized within ~ 0.3 ps and its polarization does not reverse by the negative electric fields from 0.25 ps to 0.75 ps. Given that SHG due to CO is observed in κ -Cl and theoretically predicted in κ -Cl we tentatively assign the observed SHG to the charge disproportionation state is metastable in κ -Cl, reminiscent of CO.

For the reflectivity change, the presentation of the results and discussion is more convincing with presentations of the results, followed by an interpretation "The result is explained as follows " line 175.

Reply to comment (3) of Reviewer #2

We thank Reviewer #2 for his/her valuable comment. Following the comment of Reviewer #2, we revised the corresponding discussion as follows.

[Line 13-22 of page 6 in the revised MS]

“ $\Delta I_{\text{SHG}}(t_d)$ increases in accord to $[E_{\text{THz}}(t_d)]^2$ (the red line in Fig. 2c) without delay, suggesting that the initial response is electronic in nature. $\Delta I_{\text{SHG}}(t_d)$ remains after the electric field diminishes. The important feature is that the SHG signal does not decrease so much at the time when the terahertz electric field crosses zero and changes its sign at ~ 0.2 ps. This observation suggests that the polar state formed by the first positive peak of the terahertz electric field around the time origin is stabilized within ~ 0.3 ps and its polarization does not reverse by the negative electric fields from 0.25 ps to 0.75 ps. Considering the instability to the CO with a charge disproportionation in each dimer previously reported in κ -type ET compounds⁸⁻¹², we tentatively assign the observed SHG to the polar CO in which charge disproportionation is aligned along the electric field direction.”

Comment (4) of Reviewer #2: typos

(4)-1

A scale is missing in Fig 3d for the alpha component.

(4)-2

140-141 "These results indicate that the field-induced polar state is formed by an electric field but is unstable" This is redundant "These results indicate that the polar state is formed by an electric field but is unstable" is enough

(4)-3

105 enough long -> long enough

Reply to comment (4) of Reviewer #2

We thank Reviewer #2 for his/her careful reading of our paper. We corrected the typos in the revised MS.

In addition to the changes mentioned above, we shortened several sentences to simplify the explanations and discussions.

REVIEWERS' COMMENTS

Reviewer #2 (Remarks to the Author):

I would like to thank the authors for taking the time to answer to my comments and the ones of the other reviewer.

I am sorry for the delay in review. There are many changes in the doc and sup info.

However, I find the article to be greatly improved in explaining the novelty of the charge order induced by THz field in electronic type dielectrics. Now the work is properly presented and then I recommend it for publication in Nature Communications.